# Expressive Graph Neural Networks via Equivariant Use of Noise

Xiyuan Wang [1]    Muhan Zhang [1]

## Abstract

Expressivity has been a major focus in the design of Graph Neural Networks (GNNs), yet a significant gap persists between theoretical universal expressivity and practical performance. While many expressive GNNs are efficient and achieve strong results, they often focus on specific graph properties and lack theoretical expressivity for general graph tasks. Conversely, theoretically universal-expressive models often suffer from high computational costs or poor generalization, limiting their real-world applicability. To bridge this gap, we introduce Equivariant Noise GNNs (ENGNNs), a framework that utilizes random noise features to enhance the expressivity of GNNs. Crucially, unlike prior methods that naively use noise, we enforce equivariance to nodewise noise transformations, such as orthogonal transformations. We prove that this property reduces the model's theoretical sample complexity, thereby improving generalization. Our framework simultaneously reaches theoretical universal expressivity, maintains the linear scalability of standard Message-Passing Neural Networks in practice, and achieves performance comparable to computationally expensive, high-expressivity models. Extensive experiments confirm strong performance across node, link, subgraph, and graph-level prediction tasks, demonstrating that the equivariant use of noise provides a powerful and practical pathway for building expressive GNNs. Our code is available at https://github.com/MuLabPKU/EquivNoiseGNN.

## 1. Introduction

Graph Neural Networks (GNNs) have emerged as powerful tools for graph representation learning, with applications in areas such as natural language processing (Yao et al., 2019), bioinformatics (Fout et al., 2017), and social network analysis (Chen et al., 2018). However, popular architectures like Message Passing Neural Networks (MPNNs) (Gilmer et al., 2017) face fundamental expressivity limitations, hindering their performance on complex node-, link-, and graph-level tasks (Dwivedi et al., 2022b; Li et al., 2018; Zhang & Chen, 2018; Zhang et al., 2021). Consequently, enhancing the expressivity of GNNs has become a central focus.

Research on expressivity generally follows two paths: (1) improving general capacity for graph isomorphism testing, as seen in high-order GNNs (Morris et al., 2019; Maron et al., 2019a;b), and (2) designing models to express specific graph properties relevant to a particular task, such as methods for path/neighborhood overlap between nodes (Zhu et al., 2021b; Chamberlain et al., 2023a). Path (2) is often limited to specific tasks. Path (1) is task-agnostic but has seen limited practical adoption. Among approaches for general expressivity, augmenting nodes with random noise features is theoretically appealing, as it provides a task-agnostic way to universal expressivity (Abboud et al., 2021). However, naively using noise dramatically increases the model's input space, leading to poor generalization. Early works (Sato et al., 2021; Abboud et al., 2021) verify noise's effectiveness on synthetic datasets where expressivity is paramount, but on real-world tasks, the generalization error caused by the noise often outweighs its benefits.

To overcome this generalization challenge, we propose to enforce symmetry in the noise space. We argue that while the noise itself should be random to break graph symmetries, the function processing the noise should respect graph symmetries in general. Our key insight is that by making a GNN **invariant to a group of transformations applied to each node's noise** (e.g., orthogonal transformations or channel permutations), we can drastically reduce the sample complexity bound. As shown in Figure 1, this allows the model to use noise to distinguish non-isomorphic graphs (like a 4-cycle and a 6-cycle) while correctly assigning identical representations to symmetric nodes within a single graph—a property that naive noise models struggle to maintain.

Our solution, the Equivariant Noise GNN (ENGNN), achieves invariance through a two-stream architecture. An invariant stream, initialized with standard node features, is

[1] Institute for Artificial Intelligence, Peking University. Correspondence to: Muhan Zhang <muhan@pku.edu.cn>.

*Proceedings of the 43rd International Conference on Machine Learning*, Seoul, South Korea. PMLR 306, 2026. Copyright 2026 by the author(s).

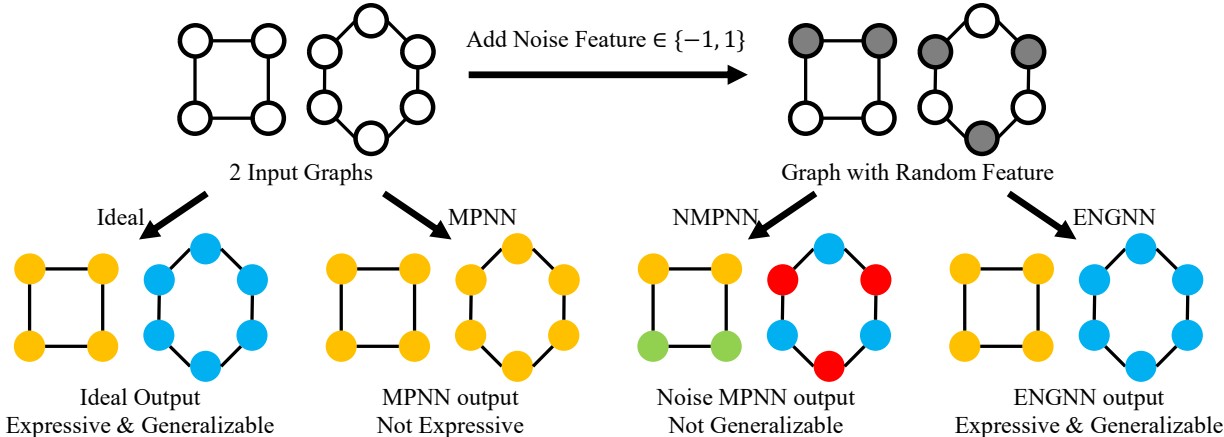

*Figure 1.* We compare vanilla MPNNs, Noise MPNNs (NMPNN), and our Equivariant Noise GNNs (ENGNN) using a 4-cycle and a 6-cycle as input, with node representations indicated by color. Ideal output should distinguish two cycles while assigning identical representations to symmetric nodes within the same cycle. MPNNs will produce the same representation to all nodes, failing to differentiate two cycles. NMPNN improves expressivity by introducing noise features (e.g., a 1D random variable in -1, 1) but compromises generalization by differentiating symmetric nodes in the same cycle. In contrast, ENGNN processes noise features equivariantly to predefined transformations (e.g. 1D orthogonal transformations), enabling it to differentiate two cycles and produce same output for symmetric nodes, thus achieving both high expressivity and better generalization than NMPNN.

processed alongside an equivariant stream, initialized with random noise. A specially designed aggregator layer, which is equivariant to a chosen set of noise transformations, then mixes information from these two streams while preserving their symmetry properties throughout the network. The final prediction relies on invariant pooling results, maintaining the output's invariance to the chosen noise transformations.

Theoretically, we establish two results: 1) We prove that ENGNNs are invariant to the chosen noise transformations, leading to a tighter generalization bound compared to naive noise methods. 2) We prove that, with a universally expressive aggregator, ENGNNs achieve universal expressivity not just for graph-level tasks, but also for node-, link-, and subgraph-level tasks, significantly broadening their applicability beyond the theory of prior noise-based GNNs.

We implement two variants, ENGNN-O (equivariant to orthogonal transformations) and ENGNN-P (equivariant to permutation), to validate our method. Experiments demonstrate that ENGNNs consistently outperform naive MPNNs and Noise MPNNs on all tasks. Furthermore, ENGNNs achieve performance comparable to computationally expensive expressive models (e.g., subgraph GNNs, high-order GNNs) while maintaining linear time and space complexity of a standard MPNN in practice. This combination of scalability, expressivity, and versatility positions ENGNN as a practical successor to MPNNs for real-world graph tasks.

## 2. Related Work

**Expressive GNNs.** Improving GNNs' expressivity is challenging due to the intricate graph topology. Research has

largely split into two directions.

The first path aims for theoretical universal expressivity: creating models that can, in principle, distinguish any two non-isomorphic graphs. These methods often draw inspiration from the Weisfeiler-Lehman (WL) test (Xu et al., 2019; Maron et al., 2019a; Morris et al., 2019; Zhang et al., 2023; Bevilacqua et al., 2022; Qian et al., 2022; Zhou et al., 2023a). Others focus on constructing function approximators for graphs, such as layers and polynomials based on high-order tensors (Maron et al., 2019b; Puny et al., 2023; Frasca et al., 2022). While theoretically powerful, these approaches often come with a high computation cost, as they rely on processing high-order tensors or sampling a large number of subgraphs (Zhang & Li, 2021; Huang et al., 2023b; Zhao et al., 2022), making them impractical for node, link, or even moderately-sized graph-level tasks. Besides high-order GNNs, another direction relaxes the strict requirement for permutation invariance by assigning nodes additional features. The relational pooling method (Murphy et al., 2019) assigns unique node IDs and pools results over different permutations, but enumerating all permutations is computationally expensive, so only a random subset is used in practice. Other methods assign nodes random noise vectors (Abboud et al., 2021; Sato et al., 2021) or integer features from heuristics (Dasoulas et al., 2020; Franks et al., 2023; Pellizzoni et al., 2024; Garg et al., 2020). While these approaches achieve universal expressivity, they often suffer from poor generalization and are not widely applied.

The second path prioritizes specific tasks over universal approximation. These models are designed to capture specific graph properties relevant to a domain. Examples include

GNNs for subgraph counting in molecules (Chen et al., 2020; Huang et al., 2023b), spectral GNNs that mimic graph filters (Wang & Zhang, 2022b; Defferrard et al., 2016; He et al., 2021; Chien et al., 2021; Klicpera et al., 2019), Graph Transformers for capturing long-range dependencies (Mialon et al., 2021; Kreuzer et al., 2021; Ying et al., 2021; Rampásek et al., 2022), positional encoding methods for improving whole graph task (Dwivedi et al., 2022a; Beaini et al., 2021; Huang et al., 2024; Wang et al., 2022; Lim et al., 2023; Huang et al., 2023a; Ma et al., 2023; Li et al., 2020), and link prediction models that leverage path information (Wang et al., 2024; Chamberlain et al., 2023a; Zhang & Chen, 2018; Yun et al., 2021; Zhu et al., 2021b). While highly effective in their target domains, these methods lack the theoretical expressivity for general graph problems.

Our work, ENGNN, bridges the gap between theoretical expressivity and real-world applicability. It proposes an equivariant use of noise to achieve universal expressivity without resorting to computationally expensive high-order operations, making it both powerful and broadly applicable.

**GNNs with Noise.** A promising strategy for increasing expressivity at a low cost is to augment nodes with random features. Initial works simply concatenate random vectors (e.g., from Gaussian or uniform distributions) to node features (Abboud et al., 2021; Sato et al., 2021). This strategy achieves universal expressivity in theory but suffers from poor generalization in practice. Resampling noise during training was proposed as a mitigation strategy (Abboud et al., 2021), but it failed to fully resolve the underlying generalization issue. Subsequent works used noise more cautiously, typically as a tool to approximate a specific heuristic. For example, MPLP (Dong et al., 2024) uses random projections to approximate common neighbor statistics, while others use noise GNNs to fit Laplacian eigenvectors or random walk encodings (Cantürk et al., 2024; Franks et al., 2025; Eliasof et al., 2023). These methods improve generalization by collapsing the noise into fixed heuristics, but they sacrifice the universal expressivity that made noise theoretically appealing. In contrast, ENGNN enforces equivariance on noise, controlling the model's sample complexity without losing the universal expressivity required for general graph tasks. Besides these works on continuous random vector features, some works (Murphy et al., 2019; Dasoulas et al., 2020) also assign random integers to nodes. Subsequent works (Franks et al., 2023; Pellizzoni et al., 2024; Garg et al., 2020) proposed assigning the same integer to some nodes to reduce sample complexity, but these works still focus on theory rather than application to broad graph tasks.

**Equivariant Graph Neural Networks.** Equivariance ensures that a model's output transforms predictably with transformations of its input. In the GNN domain, this prin-

ciple has been widely applied to physical data such as 3D molecules or point clouds. Models like TFN, EGNN, and PaiNN (Thomas et al., 2018; Satorras et al., 2021; Schütt et al., 2021; Dym & Maron, 2021; Batzner et al., 2022; Shi & Rajkumar, 2020) are designed to be equivariant to rotations and translations of the input coordinates. Here, the goal is to respect the intrinsic physical symmetries of the input data. While Satorras et al. (2021) also explore invariance in a Graph Autoencoder's latent space, the primary focus of previous equivariant GNNs remains on physical symmetries. In contrast, ENGNN does not aim to preserve a physical property of the input. It applies equivariant architectures to random noise features in order to improve the generalization of expressive noise-augmented GNNs.

## 3. Preliminaries

For a matrix $Z \in \mathbb{R}^{a \times b}$, let $Z_i \in \mathbb{R}^b$ denote the $i$-th row, $Z_{:,j} \in \mathbb{R}^a$ denote the $j$-th column, and $Z_{ij} \in \mathbb{R}$ denote the element at the $(i,j)$-th position. A *graph* is represented as $G = (V, E, X)$, where $V = 1, 2, 3, \ldots, n$ is the set of $n$ nodes, $E \subseteq V \times V$ is the set of edges, and $X \in \mathbb{R}^{n \times d}$ is the node feature matrix, with the $v$-th row $X_v$ representing the features of node $v$. $E$ can be expressed with the adjacency matrix $A \in \mathbb{R}^{n \times n}$, where $A_{uv} = 1$ if the edge $(u, v) \in E$, and 0 otherwise. A graph $G$ can be simply denoted by the tuple $(A, X)$. Let $\mathcal{G}$ denote the graph space.

**Graph Isomorphism.** A graph's structure is independent of the ordering of its nodes. This concept is formalized by *graph isomorphism*. Two graphs, $G_1 = (A_1, X_1)$ and $G_2 = (A_2, X_2)$, are *isomorphic* if there exists a *permutation* matrix $P$ such that $PA_1P^T = A_2$ and $PX_1 = X_2$.

**Message Passing Neural Network (MPNN) (Gilmer et al., 2017).** MPNN is a popular GNN framework. It consists of multiple message-passing layers, where the $k$-th layer is:

$$\boldsymbol{h}_v^{(k)} = U^{(k)}(\boldsymbol{h}_v^{(k-1)}, \mathrm{AGG}(\{M^{(k)}(\boldsymbol{h}_u^{(k-1)}) \mid (u,v) \in E\})), \quad (1)$$

where $\boldsymbol{h}_v^{(k)}$ is the representation of node $v$ at the $k$-th layer, $U^{(k)}$ and $M^{(k)}$ are functions such as Multi-Layer Perceptrons (MLPs), and AGG is an aggregation function like sum or max. The initial node representation $\boldsymbol{h}_v^{(0)}$ is the node feature $X_v$. Each layer aggregates information from neighbors to update the center node's representation.

**Equivariance and Invariance.** Given a function $h : \mathcal{X} \to \mathcal{Y}$ and a group of operators $T$ acting on $\mathcal{X}$ and $\mathcal{Y}$ through operation $\star$, $h$ is $T$-*invariant* if $h(t \star x) = h(x), \quad \forall x \in \mathcal{X}, t \in T$, and $T$-*equivariant* if $h(t \star x) = t \star h(x), \quad \forall x \in \mathcal{X}, t \in T$. In this paper there are two different symmetries. First, ENGNN is pointwise invariant/equivariant to transformations of the noise channels for a fixed graph. Second, because the noise distribution is invariant to node reordering, the expected prediction over noise is invariant to node

permutations. We focus on the first one.

**Universal Expressivity.** Following Chen et al. (2019), a deterministic GNN $f$ is considered *universally expressive* if it is *Graph-Isomorphism-discriminating*: for any two non-isomorphic graphs $G_1, G_2$, there exists a parameterization $\theta$ such that $f_\theta(G_1) \neq f_\theta(G_2)$, while isomorphic graphs receive the same output. This is equivalent to approximating continuous permutation-invariant functions on a graph domain of fixed maximum size. For the stochastic noise-augmented setting, we apply this definition to the expected predictor $f_\theta(G) = \mathbb{E}_{Z \sim \mathcal{D}_{|V(G)|}}[\text{ENGNN}_\theta(G, Z)]$, where $\mathcal{D}_n$ is a row-exchangeable noise distribution. The expectation restores node-permutation invariance, while the random noise still provides node-discriminating information almost surely.

# 4. Equivariant Noise Graph Neural Network (ENGNN)

This section introduces our Equivariant Noise Graph Neural Network (ENGNN). The core idea is to use random noise to achieve universal expressivity while leveraging the principle of equivariance to mitigate the poor generalization typically caused by naive noise injection. The framework can be adapted to be invariant to different noise transformations via specialized aggregators (e.g., orthogonal or permutation-equivariant aggregators in Appendix G).

**Key Notations.** The input graph is $G$ with $n$ nodes and $m$ edges, with auxiliary noise $Z \in \mathcal{Z}$, where $\mathcal{Z} = \mathbb{R}^{n \times C}$ and each node $i$ has a noise vector in $\mathbb{R}^C$. We consider a nodewise transformation group $T$ acting on the noise space, where each element $t \in T$ is a function $t : \mathbb{R}^C \to \mathbb{R}^C$. The transformation $t$ is applied to $Z$ row-wise, so that $t(Z)_i = t(Z_i)$. We consider functions that are equivariant or invariant to this group of noise transformations $T$. For expressivity results, $\mathcal{Z}_{\text{reg}} \subseteq \mathcal{Z}$ denotes the regular noise configurations that satisfy the required distinctness and aggregator non-degeneracy conditions; under continuous i.i.d. noise, $\mathcal{Z}_{\text{reg}}$ has probability one. For node-permutation invariance of the stochastic model, $Z$ is drawn from a row-exchangeable distribution $\mathcal{D}_n$.

## 4.1. Equivariance Helps Generalization

As shown in Figure 1, noise can break the inductive bias that GNNs should produce the same representations for symmetric nodes, leading to poor generalization. In contrast, equivariant noise GNNs reduce such cases, leading to better generalization. This section provides a theoretical explanation based on sample complexity from PAC (Probably Approximately Correct) learning theory. An introduction to PAC learning theory, related work on generalization, and proofs are in Appendix B. Note that previous works have an-

alyzed GNN generalization bounds, but our results explain the benefit of noise equivariance in our ENGNN, which previous results cannot apply directly.

We use common settings in PAC learning theory: The prediction target lies in a compact set $\mathcal{Y} \subseteq \mathbb{R}^d$. The hypothesis class $H$ consists of functions $h : \mathcal{G} \times \mathcal{Z} \to \mathcal{Y}$ mapping graph-noise pairs to predictions. The loss function $l(y, y')$ is bounded and Lipschitz continuous with constant $C_l$. The analysis treats noise as part of the input pair sampled independently in each forward pass, and the $T$-invariance considered here is pointwise in the noise: $h(G, Z) = h(G, t(Z))$. This is separate from node-permutation invariance, which holds for the expected predictor when the noise distribution is row-exchangeable.

We introduce the covering number concept and show its connection to symmetry. Intuitively, if the model is invariant to $T$, different noise points that can be transformed into each other are mapped to the same output. Therefore, the transformations can reduce the effective distance between noise data points without changing their outputs. Let $\rho_Z$ denote a metric (e.g., Euclidean distance) on the noise space $\mathcal{Z}$. The *semi-metric on the noise space induced by $T$* is:

$$\rho_{Z,T}(Z_1, Z_2) = \inf_{t_1, t_2 \in T} \rho_Z(t_1(Z_1), t_2(Z_2)), \quad (2)$$

which reduces noise distances by finding and applying transformations. The covering number for the noise space $N(\mathcal{Z}, \rho_{Z,T}, r)$ is the minimum number of points needed so that every point in $\mathcal{Z}$ is within a distance $r$ (under the semi-metric) of one of these points.

The following theorem provides a sample complexity bound for $T$-invariant models:

**Theorem 4.1.** *Assume all $h \in H$ are $C_G$-Lipschitz in $\mathcal{G}$, $C_Z$-Lipschitz in $\mathcal{Z}$, and $T$-invariant, and the loss function is $C_l$-Lipschitz. The sample complexity for empirical risk minimization is:*

$$O\left(\frac{1}{\epsilon^2} N_{Z,T} N_G \ln N_Y + \frac{1}{\epsilon^2} \ln \frac{1}{\delta}\right), \quad (3)$$

*where 1) $\epsilon, \delta$ are the error bound and failure probability, 2) $N_{Z,T} = N(\mathcal{Z}, \rho_{Z,T}, \frac{\epsilon}{12C_l C_Z})$ is the covering number for noise space $\mathcal{Z}$ with semi-metric $\rho_{Z,T}$ induced by $T$ and radius $\frac{\epsilon}{12C_l C_Z}$, 3) $N_G$ and $N_Y$ are covering numbers for graph space $\mathcal{G}$ and output space $\mathcal{Y}$, irrelevant to $T$.*

The only term related to $T$ is $N_{Z,T}$, the covering number of the noise space. For a naive Noise MPNN (which is only invariant to the identity transformation), the covering number can be enormous. For an $n \times C$-dimensional boolean noise space, $N_Z$ can be as large as $2^{nC}$, growing exponentially with the number of nodes and noise channels. This explains why noise leads to poor generalization. Incorporating symmetries in the noise space can reduce sample complexity.

As more symmetries involves (corresponding to a larger $T$), points are closer to one another under the semi-metric $\rho_{Z,T}$. This leads to a smaller covering number:

**Proposition 4.2.** *If $T_1 \subseteq T_2$, then for all $r > 0$, $N(\mathcal{Z}, \rho_{Z,T_1}, r) \geq N(\mathcal{Z}, \rho_{Z,T_2}, r)$.*

Moreover, this reduction can be dramatic. For example, by enforcing invariance to the permutation of noise channels, we can reduce the covering number by a factorial factor:

**Proposition 4.3.** *If $\mathcal{Z} = [0,1]^{n \times C}$, and $T$ includes all permutations of the $C$ noise channels, then for a small enough radius $r$: $N(\mathcal{Z}, \rho_{Z,T}, r)/N(\mathcal{Z}, \rho_Z, r) \leq 2/C!$.*

This theoretical framework shows that a principled application of invariance is key to harnessing the expressive power of noise without suffering from poor generalization. The bound is mainly qualitative rather than a tight numerical predictor: enlarging $T$ reduces $N_{Z,T}$, so the identity-only NMPNN has the loosest bound, the permutation-equivariant ENGNN-P has a tighter one, and the orthogonal-equivariant ENGNN-O can be tighter still. This trend is consistent with our empirical results. Other methods, like simply reducing the noise dimension, can hurt expressivity by causing collisions (distinct nodes receiving similar noise vectors, as shown in Appendix H and C), making invariance the superior approach.

### 4.2. Architecture

The ENGNN architecture is designed to process information while respecting these noise symmetries. Let $d$ and $L$ denote hidden dimensions, and $C$ denote noise channels. Each node $i$ maintains two representations that are updated across $K$ message-passing layers. At $k$-th layer:

- Invariant representation $X_i^{(k)} \in \mathbb{R}^d$, which remains unchanged under noise transformations.
- Equivariant representation $Z_i^{(k)} \in \mathbb{R}^{L \times C}$, which transforms as the input noise $Z_i^{(0)} \in \mathbb{R}^C$.

Initialized with noise and node features, ENGNN updates both via equivariant MPNN layers.

**Equivariant Aggregator.** An equivariant aggregator takes a multiset of invariant–equivariant feature pairs $\{(X_i^{(k)}, Z_i^{(k)}) \mid i = 1, 2, ..., B\}$ and produces a feature pair $(X', Z')$. We use AGGR to represent an aggregator. Its defining property is that, for every $t \in T$, if $(X', Z') = \text{AGGR}(\{(X_i, Z_i)\})$, then

$$\text{AGGR}(\{(X_i, t(Z_i))\}) = (X', t(Z')). \quad (4)$$

The tensor shape $Z_i^{(k)} \in \mathbb{R}^{L \times C}$ is used for all $k > 0$ so that the $C$ noise channels can transform consistently across layers, while $L$ acts as a learnable hidden feature dimension. Designs of aggregators equivariant to different transformation sets are in Appendix G. They achieve equivariance to

input, theoretical universal expressivity under mild conditions, and linear time and space complexity in the input set size in practice.

**Message-Passing Layer.** Each node $i$ updates its representations as follows:

$$
\begin{aligned}
X_i^{(k)}, Z_i^{(k)} = &\text{AGGR}_1^{(k)}\Big(\big\{\big( \\
&\text{AGGR}_2^{(k)}(\{(X_j^{(k-1)}, Z_j^{(k-1)}) \mid j \in N(i)\}, \quad (5) \\
&\text{MLP}^{(k)}(X_i^{(k-1)}), Z_i^{(k-1)})\big)\big\}\Big),
\end{aligned}
$$

where $\text{AGGR}_2^{(k)}$ aggregates neighbors' feature, $\text{AGGR}_1^{(k)}$ combines aggregated features with the center node's feature, and $\text{MLP}^{(k)}$ transforms center node's feature to distinguish it from neighbors'.

**Pooling Layer.** To generate graph-level representations, we aggregate all nodes:

$$h_G, Z_G = \text{AGGR}(\{(X_i^{(K)}, Z_i^{(K)}) \mid i \in V\}), \quad (6)$$

where the invariant output $h_G$ is used as the graph representation for downstream tasks.

For tasks involving nodes, links, or subgraphs, representations for a node subset $U \subseteq V$ is:

$$h'_U, Z'_U = \text{AGGR}_1(\{(X_i^{(K)}, Z_i^{(K)}) \mid i \in U\}), \quad (7)$$

$$h_U, Z_U = \text{AGGR}_2(\{(\text{MLP}(h_G), Z_G), (h'_U, Z'_U)\}), \quad (8)$$

where $h'_U, Z'_U$ aggregates subset node features, and $\text{AGGR}_2$ combines subset feature and global feature $h_G, Z_G$ leads to the final subgraph representations $h_U$.

**Complexity.** With efficient aggregators (Appendix G) that scale linearly with input size, ENGNN achieves $O(n + m)$ time and space complexity per message-passing layer for fixed depth, hidden dimensions, and noise channels, where $n$ is the number of nodes and $m$ the number of edges. The pooling step costs $O(n)$ time and space for full graphs and $O(n + \sum_i^B |U_i|)$ time and space for $B$ node subsets $U_1, U_2, \ldots, U_B$. Therefore, ENGNN maintains the same asymptotic scalability as vanilla MPNNs and scales much better than high-order GNNs. **Note that retaining theoretical universal expressivity may require depth and width that grow with the target graph domain, which can make the theoretical cost non-linear with respect to graph size. However, in experiments, only modest depth and width are sufficient for strong empirical performance, leading to linear time and space complexity in practice.**

### 4.3. Theoretical Expressivity

All proofs in this section are in Appendix D. First, ENGNN inherits the equivariance of its aggregator and ensures output's invariance to noise transformations:

*Table 1.* Roc-auc score ↑ of ENGNN and rGIN on synthetic and TU dataset.

| dataset | TRI(N) | TRI(X) | LCC(N) | LCC(X) | MDS(N) | MDS(X) | MUTAG | NCI1 | PROTEINS |
|---|---|---|---|---|---|---|---|---|---|
| GINs | 0.500 | 0.500 | 0.500 | 0.500 | 0.500 | 0.500 | $0.946_{\pm 0.034}$ | $0.870_{\pm 0.009}$ | $0.806_{\pm 0.029}$ |
| rGINs | 0.908 | 0.926 | 0.811 | 0.852 | 0.807 | 0.810 | $0.949_{\pm 0.040}$ | $0.876_{\pm 0.010}$ | $0.810_{\pm 0.020}$ |
| MPNN | 0.500 | 0.500 | 0.500 | 0.500 | 0.500 | 0.500 | $0.954_{\pm 0.007}$ | $0.892_{\pm 0.005}$ | $0.831_{\pm 0.037}$ |
| NMPNN | 1.000 | 1.000 | 1.000 | 1.000 | 0.933 | 0.932 | $0.972_{\pm 0.054}$ | $0.882_{\pm 0.008}$ | $0.827_{\pm 0.028}$ |
| ENGNN-P | **1.000** | **1.000** | **1.000** | **1.000** | 0.936 | 0.934 | $0.990_{\pm 0.019}$ | $0.897_{\pm 0.013}$ | $0.837_{\pm 0.027}$ |
| ENGNN-O | **1.000** | **1.000** | **1.000** | **1.000** | **0.938** | **0.939** | $\mathbf{0.991_{\pm 0.014}}$ | $\mathbf{0.902_{\pm 0.022}}$ | $\mathbf{0.843_{\pm 0.028}}$ |

*Table 2.* Mean absolute error on substructures counting. The colored cell means an error ≤ 0.01.

| Method | 3-Cyc. | 4-Cyc. | 5-Cyc. | 6-Cyc. | Tail Tri | Chor Cyc | 4-Cliq. | 4-Path | Tri-Rect |
|---|---|---|---|---|---|---|---|---|---|
| GIN | 0.3515 | 0.2742 | 0.2088 | 0.1555 | 0.3631 | 0.3114 | 0.1645 | 0.1592 | 0.2979 |
| NGNN | 0.0003 | 0.0013 | 0.0402 | 0.0439 | 0.1044 | 0.0392 | 0.0045 | 0.0244 | 0.0729 |
| GNNAK+ | 0.0004 | 0.0041 | 0.0133 | 0.0238 | 0.0043 | 0.0112 | 0.0049 | 0.0075 | 0.1311 |
| PPGN | 0.0003 | 0.0009 | 0.0036 | 0.0071 | 0.0026 | 0.0015 | 0.1646 | 0.0041 | 0.0144 |
| I2GNN | 0.0003 | 0.0016 | 0.0028 | 0.0082 | 0.0011 | 0.0010 | 0.0003 | 0.0041 | 0.0013 |
| DRFWL | 0.0004 | 0.0015 | 0.0034 | 0.0087 | 0.0030 | 0.0026 | 0.0009 | 0.0081 | 0.0070 |
| MPNN | 0.1960 | 0.1808 | 0.1658 | 0.1313 | 0.1585 | 0.1294 | 0.0598 | 0.0594 | 0.1400 |
| NMPNN | 0.0031 | 0.0121 | 0.0167 | 0.0228 | 0.0182 | 0.0179 | 0.0128 | 0.0168 | 0.0572 |
| ENGNN-P | 0.0030 | 0.0047 | 0.0058 | 0.0078 | 0.0038 | 0.0031 | 0.0016 | 0.0033 | 0.0065 |
| ENGNN-O | 0.0031 | 0.0062 | 0.0087 | 0.0092 | 0.0093 | 0.0065 | 0.0023 | 0.0099 | 0.0192 |

**Theorem 4.4.** *(Invariance) If the aggregator is equivariant to a group of nodewise noise transformations $T$, then ENGNN's outputs for graphs and subsets are $T$-invariant.*

Second, ENGNN can achieve universal expressivity in both graph and subgraph tasks:

**Theorem 4.5.** *Fix a graph domain with at most $n_{\max}$ nodes. Assume the noise distribution is continuous and row-exchangeable, and the aggregator in ENGNN achieves universal expressivity on the regular noise space $\mathcal{Z}_{\mathrm{reg}}$ (e.g., the aggregators in Appendix G).*

*(Graph-Level Expressivity) Let $ENGNN_\theta(G, Z)$ denote the model output for graph $G$ and noise $Z$. For any non-isomorphic graphs $G$ and $H$ in this fixed-size domain, there exists a parameterization $\theta$ such that their possible outputs under regular noise are disjoint:*

$$ENGNN(G, Z_1) \neq ENGNN(H, Z_2), \quad \forall Z_1, Z_2 \in \mathcal{Z}_{\mathrm{reg}}. \quad (9)$$

*Consequently, the expected predictor $f_\theta(G) = \mathbb{E}_{Z \sim \mathcal{D}_{|V(G)|}}[ENGNN_\theta(G, Z)]$ is graph-isomorphism-discriminating.*

*(Subgraph-Level Expressivity) Let $ENGNN_\theta(U, G, Z)$ denote the output for subset $U$ in graph $G$. For any non-isomorphic pairs $(G, U_G)$ and $(H, U_H)$ in the same fixed-size domain, there exists a parameterization $\theta$ such that the corresponding output sets under $\mathcal{Z}_{\mathrm{reg}}$ are disjoint.*

$$ENGNN(U_G, G, Z_1) \neq ENGNN(U_H, H, Z_2), \forall Z_1, Z_2 \in \mathcal{Z}_{\mathrm{reg}}. \quad (10)$$

*Hence the expected subset predictor is also isomorphism-discriminating.*

These results demonstrate that ENGNN can distinguish non-isomorphic graphs and their subsets under suitable parameterizations on graph spaces of fixed maximum size, making it theoretically expressive for general graph tasks.

## 5. Experiments

In this section, we conduct a comprehensive empirical evaluation of ENGNN across graph, node, link, and subgraph-level tasks to demonstrate its broad effectiveness and scalability. As prior work has often focused on only one of these tasks, we use different relevant baselines for each task type.

We evaluate our two primary variants: ENGNN-P (using a permutation-equivariant aggregator, see Appendix G for details) and ENGNN-O (using an orthogonal-transformation-equivariant aggregator). All ENGNN models use noise features sampled i.i.d. from a normal distribution, and the noise is resampled in each forward pass following previous work (Abboud et al., 2021). This implements the stochastic predictor discussed in Section 3; empirically, a single sampled forward pass is used unless otherwise stated. For ablation studies, we compare against a vanilla MPNN (without noise) and a Noise MPNN (NMPNN), which represents the inequivariant noise approach from prior work. All models are trained in a supervised manner. As our main focus is real-world performance, experiments verifying generalization gain are in Appendix I. Detailed experimental settings for ENGNN, its ablation variants, and the baselines can be found in Appendix E. Dataset statistics and splits are

*Table 3.* Graph property prediction Results.

|  | zinc MAE↓ | zinc-full MAE↓ | molhiv AUC↑ |
|---|---|---|---|
| GIN | $0.163_{\pm0.004}$ | $0.088_{\pm0.002}$ | $77.07_{\pm1.49}$ |
| Graphormer | $0.122_{\pm0.006}$ | - | $\mathbf{80.51_{\pm0.53}}$ |
| GPS | $\mathbf{0.070_{\pm0.004}}$ | - | $78.80_{\pm1.01}$ |
| NGNN | $0.111_{\pm0.003}$ | $0.029_{\pm0.001}$ | $78.34_{\pm1.86}$ |
| GNNAK+ | $0.080_{\pm0.001}$ | – | $79.61_{\pm1.19}$ |
| PPGN | $0.079_{\pm0.005}$ | $0.022_{\pm0.003}$ | - |
| I2GNN | $0.083_{\pm0.001}$ | $0.023_{\pm0.002}$ | $78.68_{\pm0.93}$ |
| DRFWL | $0.077_{\pm0.002}$ | $0.025_{\pm0.003}$ | $78.18_{\pm2.19}$ |
| MPNN | $0.131_{\pm0.007}$ | $0.046_{\pm0.002}$ | $78.27_{\pm1.14}$ |
| NMPNN | $0.136_{\pm0.007}$ | $0.051_{\pm0.004}$ | $77.74_{\pm0.98}$ |
| ENGNN-P | $0.091_{\pm0.005}$ | $0.026_{\pm0.003}$ | $78.51_{\pm0.86}$ |
| ENGNN-O | $\mathbf{0.070_{\pm0.006}}$ | $\mathbf{0.022_{\pm0.003}}$ | $78.63_{\pm0.93}$ |

presented in Appendix F.

**Whole Graph Tasks.** We evaluate ENGNN on graph-level tasks. First, we compare our ENGNN with naive noise MPNNs, specifically rGIN (Sato et al., 2021), on six synthetic and three TU datasets (Ivanov et al., 2019). As shown in Table 1, both ENGNN-P and ENGNN-O consistently outperform all baselines across all tasks, showing that **ENGNN significantly outperforms vanilla noise methods**. We also include other noise methods as baselines CLIP (Dasoulas et al., 2020), RP (Murphy et al., 2019), IRNI (Franks et al., 2023), and GPSE (Franks et al., 2025; Eliasof et al., 2023); ENGNN still outperforms these noise methods. The results are shown in Appendix I. We additionally include Graph Transformer baselines in Table 3 and BREC expressivity results in Appendix I.

Next, we benchmark ENGNN against computationally expensive, highly expressive models, including NGNN (Zhang & Li, 2021), GNNAK+ (Zhao et al., 2022), I2GNN (Huang et al., 2023b), PPGN (Maron et al., 2019a), DRFWL (Zhou et al., 2023b), as well as the expressive MPNN variant GIN (Xu et al., 2019). We evaluate performance on a subgraph counting task, where the goal is to regress the number of occurrences of various subgraphs, and on three graph property prediction datasets: zinc, zinc-full (Gómez-Bombarelli et al., 2016), and ogbg-molhiv (Hu et al., 2020).

For the subgraph counting task, we follow the setup in Zhou et al. (2023b), where a model is considered capable of counting a subgraph if it achieves a loss lower than 0.01. Results are shown in Table 2. The evaluated subgraphs include 3–6-Cyc (cycles of length 3 to 6), Tail-Tri (Tailed Triangle), Chor-Cyc (cycle with a chord), 4-Cliq (4-Clique), 4-Path (path of length 4), and Tri-Rect (a triangle connected to a rectangle). Vanilla MPNN fails to count any complex subgraphs. In contrast, ENGNN-P successfully counts all target subgraphs, and ENGNN-O performs competitively. Notably, while NMPNN fails to meet the success threshold, it still performs far better than the MPNN, confirming that **equivariant use of noise provides a effective expressivity**

**boost**.

The results on graph property prediction datasets are in Table 3. On the ZINC and ZINC-full datasets, ENGNN-O achieves the best or tied-best performance, while ENGNN-P also achieves competitive performance. Compared with Graph Transformers, ENGNN-O matches GPS on ZINC and remains competitive on MOLHIV. On MOLHIV, ENGNN is outperformed by Graphormer, GNNAK+, and I2GNN; this may be attributed to the strong inductive bias of specialist architectures. In particular, molecular graphs often contain hierarchical structures such as atoms, functional groups, and whole molecules, and subgraph-based GNNs explicitly encode such local substructures. Nevertheless, ENGNN still achieves strong results on this task. Notably, ENGNN is significantly more scalable than both high-order and subgraph GNNs, requiring as little as 10% of the time and GPU memory compared to subgraph GNNs, as shown in Table 7. **These results demonstrate that ENGNN achieves performance comparable to powerful specialist models at a fraction of the computational cost.**

**Node Tasks.** We evaluate our models on real-world node classification tasks. Following previous work (Chien et al., 2021), we use 8 node classification datasets including 5 homogeneous graphs Cora, CiteSeer, PubMed (Yang et al., 2016), Photo, and Amazon (Shchur et al., 2018), and 3 heterogeneous graphs Chameleon, Squirrel (Rozemberczki et al., 2021), and Actor (Pei et al., 2020). Our baselines includes widely used node classification GNNs: GCN (Kipf & Welling, 2016), APPNP (Klicpera et al., 2019), ChebyNet (Defferrard et al., 2016), GPRGNN (Chien et al., 2021), and BernNet (He et al., 2021). The experimental results are presented in Table 4. ENGNN surpasses all baselines on 7/8 datasets, showing its **strong capacity for node tasks**.

**Link Tasks.** We evaluate our models on link prediction datasets, including three citation graphs (Yang et al., 2016) (Cora, Citeseer, and Pubmed) and three Open Graph Benchmark (Hu et al., 2020) datasets (Collab, PPA, and DDI). We employ a range of baseline methods, encompassing traditional heuristics like CN (Barabási & Albert, 1999), RA (Zhou et al., 2009), and AA (Adamic & Adar, 2003), as well as GAE models, such as GCN (Kipf & Welling, 2016) and SAGE (Hamilton et al., 2017). Additionally, we consider models involving pairwise representations, including SEAL (Zhang & Chen, 2018) and NBFNet (Zhu et al., 2021b), as well as SF-and-MPNN models like Neo-GNN (Yun et al., 2021) and BUDDY (Chamberlain et al., 2023b). The baseline results are sourced from (Chamberlain et al., 2023b). The experimental results are presented in Table 5. **Our ENGNN achieves best or second-best performance on 5 out of 6 evaluated datasets.**

**Subgraph Tasks.** We evaluate our models on subgraph clas-

*Table 4.* Results on node classification datasets: Mean accuracy (%) $\pm$ standard variation.

| Dataset | Cora | Citeseer | Pubmed | Computers | Photo | Chameleon | Actor | Squirrel |
|---|---|---|---|---|---|---|---|---|
| GCN | $87.14_{\pm1.01}$ | $79.86_{\pm0.67}$ | $86.74_{\pm0.27}$ | $83.32_{\pm0.33}$ | $88.26_{\pm0.73}$ | $59.61_{\pm2.21}$ | $33.23_{\pm1.16}$ | $46.78_{\pm0.87}$ |
| GIN | $86.58_{\pm0.97}$ | $77.11_{\pm0.76}$ | $86.93_{\pm0.26}$ | $58.87_{\pm7.55}$ | $87.13_{\pm4.52}$ | $66.87_{\pm2.72}$ | $36.66_{\pm7.53}$ | $40.53_{\pm1.16}$ |
| GAT | $88.03_{\pm0.79}$ | $80.52_{\pm0.71}$ | $87.04_{\pm0.24}$ | $83.23_{\pm0.39}$ | $90.94_{\pm0.68}$ | $63.13_{\pm1.93}$ | $33.93_{\pm2.47}$ | $44.49_{\pm0.88}$ |
| APPNP | $88.14_{\pm0.73}$ | $\mathbf{80.47_{\pm0.74}}$ | $88.12_{\pm0.31}$ | $85.32_{\pm0.37}$ | $88.51_{\pm0.31}$ | $51.84_{\pm1.82}$ | $39.66_{\pm0.55}$ | $34.71_{\pm0.57}$ |
| ChebyNet | $86.67_{\pm0.82}$ | $79.11_{\pm0.75}$ | $87.95_{\pm0.28}$ | $87.54_{\pm0.43}$ | $93.77_{\pm0.32}$ | $59.28_{\pm1.25}$ | $37.61_{\pm0.89}$ | $40.55_{\pm0.42}$ |
| GPRGNN | $88.57_{\pm0.69}$ | $80.12_{\pm0.83}$ | $88.46_{\pm0.33}$ | $86.85_{\pm0.25}$ | $93.85_{\pm0.28}$ | $67.28_{\pm1.09}$ | $39.92_{\pm0.67}$ | $50.15_{\pm1.92}$ |
| BernNet | $88.52_{\pm0.95}$ | $80.09_{\pm0.79}$ | $88.48_{\pm0.41}$ | $87.64_{\pm0.44}$ | $93.63_{\pm0.35}$ | $68.29_{\pm1.58}$ | $41.79_{\pm1.01}$ | $51.35_{\pm0.73}$ |
| MPNN | $87.36_{\pm0.52}$ | $79.62_{\pm0.75}$ | $89.53_{\pm0.29}$ | $89.53_{\pm0.83}$ | $94.74_{\pm0.25}$ | $67.18_{\pm1.07}$ | $40.41_{\pm1.53}$ | $51.99_{\pm1.78}$ |
| NMPNN | $20.11_{\pm2.01}$ | $20.80_{\pm2.63}$ | $69.28_{\pm3.14}$ | $66.42_{\pm1.39}$ | $65.12_{\pm1.95}$ | $41.25_{\pm1.38}$ | $23.73_{\pm2.36}$ | $38.25_{\pm1.04}$ |
| ENGNN-P | $88.85_{\pm0.96}$ | $79.97_{\pm0.79}$ | $\mathbf{89.79_{\pm0.64}}$ | $\mathbf{90.48_{\pm0.31}}$ | $\mathbf{95.24_{\pm0.58}}$ | $71.40_{\pm1.29}$ | $40.64_{\pm0.67}$ | $52.77_{\pm1.43}$ |
| ENGNN-O | $\mathbf{89.32_{\pm1.66}}$ | $79.67_{\pm0.70}$ | $89.32_{\pm0.50}$ | $87.96_{\pm0.91}$ | $94.00_{\pm0.80}$ | $\mathbf{71.51_{\pm2.51}}$ | $\mathbf{45.76_{\pm1.85}}$ | $\mathbf{64.66_{\pm1.25}}$ |

*Table 5.* Results on link prediction benchmarks. OOM means out of GPU memory.

| Metric | Cora HR@100 | Citeseer HR@100 | Pubmed HR@100 | Collab HR@50 | PPA HR@100 | DDI HR@20 |
|---|---|---|---|---|---|---|
| CN | $33.92_{\pm0.46}$ | $29.79_{\pm0.90}$ | $23.13_{\pm0.15}$ | $56.44_{\pm0.00}$ | $27.65_{\pm0.00}$ | $17.73_{\pm0.00}$ |
| AA | $39.85_{\pm1.34}$ | $35.19_{\pm1.33}$ | $27.38_{\pm0.11}$ | $64.35_{\pm0.00}$ | $32.45_{\pm0.00}$ | $18.61_{\pm0.00}$ |
| RA | $41.07_{\pm0.48}$ | $33.56_{\pm0.17}$ | $27.03_{\pm0.35}$ | $64.00_{\pm0.00}$ | $49.33_{\pm0.00}$ | $27.60_{\pm0.00}$ |
| GCN | $66.79_{\pm1.65}$ | $67.08_{\pm2.94}$ | $53.02_{\pm1.39}$ | $44.75_{\pm1.07}$ | $18.67_{\pm1.32}$ | $37.07_{\pm5.07}$ |
| SAGE | $55.02_{\pm4.03}$ | $57.01_{\pm3.74}$ | $39.66_{\pm0.72}$ | $48.10_{\pm0.81}$ | $16.55_{\pm2.40}$ | $53.90_{\pm4.74}$ |
| SEAL | $81.71_{\pm1.30}$ | $83.89_{\pm2.15}$ | $75.54_{\pm1.32}$ | $64.74_{\pm0.43}$ | $48.80_{\pm3.16}$ | $30.56_{\pm3.86}$ |
| NBFnet | $71.65_{\pm2.27}$ | $74.07_{\pm1.75}$ | $58.73_{\pm1.99}$ | OOM | OOM | $4.00_{\pm0.58}$ |
| Neo-GNN | $80.42_{\pm1.31}$ | $84.67_{\pm2.16}$ | $73.93_{\pm1.19}$ | $57.52_{\pm0.37}$ | $\underline{49.13_{\pm0.60}}$ | $63.57_{\pm3.52}$ |
| BUDDY | $\underline{88.00_{\pm0.44}}$ | $\mathbf{92.93_{\pm0.27}}$ | $74.10_{\pm0.78}$ | $\mathbf{65.94_{\pm0.58}}$ | $\mathbf{49.85_{\pm0.20}}$ | $\mathbf{78.51_{\pm1.36}}$ |
| MPNN | $86.26_{\pm1.64}$ | $90.40_{\pm1.71}$ | $79.48_{\pm3.74}$ | $62.84_{\pm1.07}$ | $5.62_{\pm2.52}$ | $24.76_{\pm15.29}$ |
| NMPNN | $48.12_{\pm11.94}$ | $68.63_{\pm7.29}$ | $63.96_{\pm1.92}$ | $7.35_{\pm7.04}$ | $39.90_{\pm5.52}$ | $23.08_{\pm5.89}$ |
| ENGNN-P | $\mathbf{88.10_{\pm1.67}}$ | $\underline{91.56_{\pm1.02}}$ | $81.26_{\pm1.20}$ | $63.69_{\pm0.82}$ | $44.97_{\pm0.74}$ | $27.64_{\pm6.21}$ |
| ENGNN-O | $87.96_{\pm1.63}$ | $88.12_{\pm0.97}$ | $\mathbf{82.08_{\pm2.16}}$ | $\underline{65.34_{\pm0.45}}$ | $48.44_{\pm1.93}$ | $\underline{77.61_{\pm4.50}}$ |

sification tasks. Datasets include three synthetic datasets: density, cut ratio, coreness, and four real-world subgraph datasets, namely ppi-bp, em-user, hpo-metab, hpo-neuro (Alsentzer et al., 2020). We consider three baselines: SubGNN (Alsentzer et al., 2020) with subgraph-level message passing, Sub2Vec (Adhikari et al., 2018) sampling random walks in subgraphs and encoding them with RNN, GLASS (Wang & Zhang, 2022a) using MPNN with labeling trick. The results are shown in Table 6. **Our ENGNN achieves best performance on 4/7 datasets and second best performance on 3/7 datasets.**

**Scalability.** We present the training time per epoch, GPU memory consumption, and training loss curves in Table 7. Our ENGNN-O achieves comparable resource comsumption to simple MPNN method and takes much less time and memory than high-order GNNs.

**Summary of Experiments.** Across diverse graph, node, link, and subgraph-level tasks, ENGNN demonstrates highly competitive performance, often exceeding that of models specifically designed for a single task type. The comprehensive ablation studies confirm our core hypothesis: ENGNN consistently outperforms vanilla MPNNs and naive Noise MPNNs, highlighting the potential of equivariant noise.

## 6. Conclusion

To bridge the gap between real-world applicability and theoretical universal expressivity, we propose equivariant noise GNN. It that utilize noise equivariantly for better generalization bound. Our approach demonstrates universal theoretical expressivity and excels in real-world performance. It extends the design space of GNN and provides a principled way to utilize noise feature.

*Table 6.* Mean Micro-F1 with standard error of the mean on subgraph tasks.

| Method | density | cut ratio | coreness | ppi-bp | hpo-metab | hpo-neuro | em-user |
|---|---|---|---|---|---|---|---|
| GLASS | $0.930_{\pm 0.009}$ | $0.935_{\pm 0.006}$ | $0.840_{\pm 0.009}$ | $\mathbf{0.619_{\pm 0.007}}$ | $\mathbf{0.614_{\pm 0.005}}$ | $\mathbf{0.685_{\pm 0.005}}$ | $0.888_{\pm 0.006}$ |
| SubGNN | $0.919_{\pm 0.006}$ | $0.629_{\pm 0.013}$ | $0.659_{\pm 0.031}$ | $0.599_{\pm 0.008}$ | $0.537_{\pm 0.008}$ | $0.644_{\pm 0.006}$ | $0.816_{\pm 0.013}$ |
| Sub2Vec | $0.459_{\pm 0.012}$ | $0.354_{\pm 0.014}$ | $0.360_{\pm 0.019}$ | $0.388_{\pm 0.001}$ | $0.472_{\pm 0.010}$ | $0.618_{\pm 0.003}$ | $0.779_{\pm 0.013}$ |
| MPNN | $0.321_{\pm 0.023}$ | $0.311_{\pm 0.012}$ | $0.545_{\pm 0.024}$ | $0.547_{\pm 0.009}$ | $0.500_{\pm 0.010}$ | $0.587_{\pm 0.004}$ | $0.641_{\pm 0.017}$ |
| NMPNN | $0.321_{\pm 0.023}$ | $0.311_{\pm 0.012}$ | $0.527_{\pm 0.016}$ | $0.516_{\pm 0.010}$ | $0.460_{\pm 0.010}$ | $0.582_{\pm 0.006}$ | $0.896_{\pm 0.006}$ |
| ENGNN-P | $0.572_{\pm 0.021}$ | $0.744_{\pm 0.050}$ | $0.742_{\pm 0.014}$ | $0.581_{\pm 0.007}$ | $0.540_{\pm 0.008}$ | $0.590_{\pm 0.003}$ | $\mathbf{0.902_{\pm 0.006}}$ |
| ENGNN-O | $\mathbf{0.992_{\pm 0.003}}$ | $\mathbf{0.984_{\pm 0.007}}$ | $\mathbf{0.842_{\pm 0.026}}$ | $0.607_{\pm 0.003}$ | $0.573_{\pm 0.004}$ | $0.579_{\pm 0.006}$ | $0.847_{\pm 0.017}$ |

*Table 7.* Time (s) per epoch and GPU memory (GB) consumption on zinc with batch size 128.

| | MPNN | ENGNN-O | SUN | SSWL | PPGN |
|---|---|---|---|---|---|
| Time/s | 2.36 | 4.81 | 20.93 | 45.30 | 20.21 |
| Memory/GB | 0.24 | 0.62 | 3.72 | 3.89 | 20.37 |

## 7. Limitations

Although the ENGNN introduced in our work shares the same time complexity as traditional MPNNs, the inclusion of noise features introduces additional computational overhead. Furthermore, despite the fact that noise is not task-specific, our approach requires modifications to the aggregator, which means it cannot be seamlessly integrated with other existing GNNs. Future work will focus on further reducing computational complexity and developing a plug-and-play method.

## Impact Statement

This paper presents work whose goal is to advance the field of graph representation learning and will improve the design of graph generation and prediction models. There are many potential societal consequences of graph learning improvement, such as accelerating drug discovery, improving traffic efficiency, and providing better recommendation in social network. None of them we feel need to be specifically highlighted here for potential risk.

## Acknowledgement

This work is supported by National Natural Science Foundation of China (62550138, 62276003).

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

## A. The Use of Large Language Models (LLMs)

We use Gemini 2.5 flash to proofread our writing in our paper. We also use it to proofread our proof and keep notations consistent.

## B. Proofs for Section 4.1

PAC (Probably Approximately Correct) learning theory is widely used for analyzing generalization. Previous works have analyse the generalization of GNNs (Morris et al., 2023; Pellizzoni et al., 2024; Garg et al., 2020), but there results cannot directly apply to our case. Previous works (Alberto et al., 2021; Behrooz & Jegelka, 2023) using PAC to analyze that symmetry can improve the generalization of kernel regression. Elesedy (2022); Mroueh et al. (2015); Sannai et al. (2021); Zhu et al. (2021a) show that symmetry can improve generalization for neural network models. However, they primarily focus on a general learning setting. In this work, we propose to use noise invariance to boost the generalization for GNNs with random features. Some previous works (Alberto et al., 2021; Behrooz & Jegelka, 2023) also introduce noise as auxiliary features into equivariant networks for performing approximately equivariant tasks (that are not strictly equivariant). However, they are equivariant to operations on the original data, not to noise transformations. In this section, we first restate our notations, then give formal definitions for important concepts in Section 4.1, and provide proofs for conclusions in Section 4.1.

**Notations:** Consider a graph $G = (A, X) \in \mathcal{G}$ with $n$ nodes, where $A \in \mathbb{R}^{n \times n}$ is the adjacency matrix and $X \in \mathbb{R}^{n \times d}$ is the node feature matrix. Additionally, there is auxiliary feature noise $Z \in \mathcal{Z} = \mathbb{R}^{n \times C}$. The input domain is $\mathcal{I} = G \times \mathcal{Z}$, and the target domain is $\mathcal{Y} \subseteq \mathbb{R}^{d'}$. We introduce a transformation set $T$. For an operation $t \in T$ on the noise, the group acts on the data $(A, X, Z)$ as $(A, X, t(Z))$.

A learning task is defined as $(I, Y, l)$, where $I$ and $Y$ are random elements in $\mathcal{I}$ and $\mathcal{Y}$, respectively, and $l : \mathcal{Y} \times \mathcal{Y} \to \mathbb{R}^+$ is an integrable, bounded, and $C_l$-Lipschitz loss function. Let $\mathcal{H}$ be a class of measurable functions $h : \mathcal{I} \to \mathcal{Y}$, known as the hypothesis class. An algorithm $\text{alg} : \bigcup_{i \in \mathbb{N}} (\mathcal{I} \times \mathcal{Y})^i \to \mathcal{H}$ maps a finite sequence of data points to a hypothesis in $\mathcal{H}$. In our work, we assume the algorithm is $T$-invariant. For example, when $\mathcal{H}$ consists of invariant functions, empirical risk minimization is $T$-invariant.

**Definitions:**

**Definition B.1.** (Sample Complexity) Algorithm alg learns $\mathcal{H}$ with respect to a task $(I, Y, l)$ if there exists a function $m : (0, 1)^2 \to \mathbb{N}$ such that for all $\epsilon, \delta \in (0, 1)$, if $n > m(\epsilon, \delta)$, then:

$$P\left(\mathbb{E}\left[l(h_S(I), Y) \mid S\right] \geq \inf_{h \in \mathcal{H}} \mathbb{E}\left[l(h(I), Y)\right] + \epsilon\right) \leq \delta, \tag{11}$$

where $h_S = \text{alg}(S)$ and $S \sim (I, Y)^n$ is an i.i.d. sample. The *sample complexity* of alg is the minimal $m(\epsilon, \delta)$ satisfying this condition.

**Definition B.2.** (Semi-Metric) Given a set $\mathcal{X}$ and a function $\rho : \mathcal{X} \times \mathcal{X} \to \mathbb{R}$, if for all $x, y, z \in \mathcal{X}$:

- $\rho(x, y) \geq 0$,
- $\rho(x, x) = 0$,
- $\rho(x, y) = \rho(y, x)$,

then $\rho$ is a semi-metric on $\mathcal{X}$, and $(\mathcal{X}, \rho)$ is a semi-metric space. If $\rho(x, y) > 0$ for all $x \neq y$, $\rho$ is a metric.

For $\mathcal{G}$, $\mathcal{Z}$, and $\mathcal{Y}$, we define semi-metrics $\rho_G$, $\rho_Z$, and $\rho_Y$ as follows. For graph comparison, graphs in a fixed maximum-size domain can be padded with isolated zero-feature nodes to share the same matrix size.

- $\rho_G$: For graphs $G_1 = (A_1, X_1)$ and $G_2 = (A_2, X_2)$,

$$\rho_G(G_1, G_2) = \min_{P \in S_n} \left(\|PX_1 - X_2\|_1 + \|PA_1P^T - A_2\|_1\right), \tag{12}$$

where $S_n$ is the symmetric group. This version is invariant to node reordering.

- $\rho_Z$ and $\rho_Y$: $\ell_1$-norm differences:

$$\rho_Z(Z_1, Z_2) = \|Z_1 - Z_2\|_1, \quad \rho_Y(y_1, y_2) = \|y_1 - y_2\|_1. \tag{13}$$

**Definition B.3.** (Cover) A $\delta$-*cover* of a semi-metric space $(\mathcal{X}, \rho)$ is a set $S \subseteq \mathcal{X}$ such that $\forall x \in \mathcal{X}, \exists s \in S, \rho(s, x) \leq \delta$. The *covering number* $N(\mathcal{X}, \rho, r)$ is the minimal $|S|$ for radius $r$.

## B.1. Proof for Theorem 4.1

We first bound the sample complexity using the covering number of the hypothesis space.

**Lemma B.4.** *The sample complexity for an algorithm with $C_l$-Lipschitz loss function is*

$$O\left(\frac{1}{\epsilon^2}\left(\ln N\left(\mathcal{H}, \rho_H, \frac{\epsilon}{4C_l}\right) + \ln\frac{1}{\delta}\right)\right), \tag{14}$$

*where $\rho_H$ is a semi-metric on $\mathcal{H}$, defined as $\rho_H(h, h') = \sup_{I \in \mathcal{I}} \|h(I) - h'(I)\|$.*

*Proof.* Given random variables $I, Y$, define

$$L(h) = \mathbb{E}[l(h(I), Y)] - \frac{1}{n}\sum_{i=1}^{n} l(h(I_i), Y_i). \tag{15}$$

For $h, h' \in \mathcal{H}$:

$$|L(h) - L(h')| \leq \mathbb{E}\left[|l(h(I), Y) - l(h'(I), Y)|\right] + \frac{1}{n}\sum_{i=1}^{n}|l(h(I_i), Y_i) - l(h'(I_i), Y_i)| \tag{16}$$

$$\leq 2C_l\rho_H(h, h'). \tag{17}$$

Let $K$ be a $\kappa$-cover of $\mathcal{H}$, and define $D(k) = \{h \in \mathcal{H} \mid \rho_H(h, k) \leq \kappa\}$. Then:

$$\mathbb{P}\left[\sup_{h \in \mathcal{H}}|L(h)| \geq \epsilon\right] \leq \sum_{k \in K}\mathbb{P}\left[\sup_{h \in D(k)}|L(h)| \geq \epsilon\right] \tag{18}$$

$$\leq \sum_{k \in K}\mathbb{P}\left[|L(k)| + 2C_l\kappa \geq \epsilon\right] \tag{19}$$

$$\leq \sum_{k \in K}\mathbb{P}\left[|L(k)| \geq (1-\alpha)\epsilon\right], \quad \alpha = \frac{2C_l\kappa}{\epsilon}. \tag{20}$$

By Hoeffding's inequality (assuming $l(h(I), Y) \in [0, 1]$):

$$\mathbb{P}\left[|L(k)| \geq (1-\alpha)\epsilon\right] \leq 2\exp\left(-2n(1-\alpha)^2\epsilon^2\right). \tag{21}$$

For $\alpha = \frac{1}{2}$:

$$\mathbb{P}\left[\sup_{h \in \mathcal{H}}|L(h)| \geq \epsilon\right] \leq 2N\left(\mathcal{H}, \rho_H, \frac{\epsilon}{4C_l}\right)\exp\left(-\frac{n\epsilon^2}{2}\right). \tag{22}$$

Thus, the sample complexity is $O\left(\frac{1}{\epsilon^2}\left(\ln N\left(\mathcal{H}, \rho_H, \frac{\epsilon}{4C_l}\right) + \ln\frac{1}{\delta}\right)\right)$. $\square$

We further decompose the input space $\mathcal{I}$ into $G \times \mathcal{Z}$:

**Lemma B.5.** *If $\mathcal{H}$ contains functions partially Lipschitz in $G$ (constant $C_G$) and $\mathcal{Z}$ (constant $C_Z$):*

$$N(\mathcal{H}, \rho_H, r) \leq N\left(\mathcal{Y}, \rho_Y, \frac{r}{3}\right)^{N\left(G, \rho_G, \frac{r}{3C_G}\right)N\left(\mathcal{Z}, \rho_{Z,T}, \frac{r}{3C_Z}\right)}. \tag{23}$$

*Proof.* Let $I$, $J$, and $K$ be $r_1$-, $r_2$-, and $r_3$-covers of $G$, $\mathcal{Z}$, and $\mathcal{Y}$, respectively. Construct:

$$F = \left\{f_k \mid k \in K^{|I| \times |J|}, f_k(G, Z) = k_{ij} \text{ if } G \in D(I_i), Z \in D(J_j)\right\}, \tag{24}$$

where $D(I_i)$ is intuitively the set of graph close to $I_i$: $D(I_i) \subseteq G$, $D(I_i) \cap D(I_j) = $ if $i \neq j$, $\forall G \in D(I_i), \rho_G(G, I_i) \leq r_1$, and $\cup_i D(I_i) = G$. $D(J_j)$ is defined for $\mathcal{Z}$ similarly.

For all $h \in \mathcal{H}$:

$$\min_{f \in F} \rho_H(f, h) \leq \min_{f \in F} \max_{i \in I} \max_{j \in J} \sup_{G \in D(I_i)} \sup_{Z \in D(J_j)} \left( \|f(G, Z) - f(I_i, Z)\| \right. \tag{25}$$

$$+ \|f(I_i, Z) - f(I_i, J_j)\| + \|f(I_i, J_j) - h(I_i, J_j)\| \right) \tag{26}$$

$$\leq C_G r_1 + C_Z r_2 + r_3. \tag{27}$$

Setting $r_1 = \frac{r}{3C_G}$, $r_2 = \frac{r}{3C_Z}$, and $r_3 = \frac{r}{3}$, we obtain:

$$N(\mathcal{H}, \rho_H, r) \leq N\left(\mathcal{Y}, \rho_Y, \frac{r}{3}\right)^{N\left(\mathcal{G}, \rho_G, \frac{r}{3C_G}\right) N\left(\mathcal{Z}, \rho_{Z,T}, \frac{r}{3C_Z}\right)}. \tag{28}$$

$\square$

Combining Lemma B.1 and Lemma B.2 gives the sample complexity

$$O\left(\frac{1}{\epsilon^2}\left(N\left(\mathcal{G}, \rho_G, \frac{\epsilon}{12 C_l C_G}\right) N\left(\mathcal{Z}, \rho_{Z,T}, \frac{\epsilon}{12 C_l C_Z}\right) \ln N\left(\mathcal{Y}, \rho_Y, \frac{\epsilon}{12 C_l}\right) + \ln \frac{1}{\delta}\right)\right). \tag{29}$$

The graph and noise covering numbers appear outside the logarithm because Lemma B.2 covers the hypothesis class by discretizing the input domain: there are $N_G N_{Z,T}$ graph–noise cells, and each cell can be assigned one of $N_Y$ representative outputs, yielding $N(\mathcal{H}) \leq N_Y^{N_G N_{Z,T}}$ and hence $\ln N(\mathcal{H}) \leq N_G N_{Z,T} \ln N_Y$.

## B.2. Proof for Proposition 4.2

**Proposition B.6.** *If $T_1 \subseteq T_2$, then for all $r > 0$, $N(\mathcal{Z}, \rho_{Z,T_1}, r) \geq N(\mathcal{Z}, \rho_{Z,T_2}, r)$.*

*Proof.* For all $Z_1, Z_2 \in \mathcal{Z}$,

$$\rho_{Z,T_1}(Z_1, Z_2) = \inf_{t,t' \in T_1} \rho_Z(t(Z_1), t'(Z_2)) \geq \inf_{t,t' \in T_2} \rho_Z(t(Z_1), t'(Z_2)) = \rho_{Z,T_2}(Z_1, Z_2). \tag{30}$$

Let $S$ be an $r$-cover of $(\mathcal{Z}, \rho_{Z,T_1})$ with $|S| = N(\mathcal{Z}, \rho_{Z,T_1}, r)$. For any $Z \in \mathcal{Z}$, there exists $Z' \in S$ such that $\rho_{Z,T_2}(Z, Z') \leq \rho_{Z,T_1}(Z, Z') \leq r$. Thus, $S$ is also an $r$-cover for $(\mathcal{Z}, \rho_{Z,T_2})$, implying $N(\mathcal{Z}, \rho_{Z,T_2}, r) \leq N(\mathcal{Z}, \rho_{Z,T_1}, r)$. $\square$

## B.3. Proof for Proposition 4.3

**Proposition B.7.** *If $\mathcal{Z} = [0, 1]^{n \times C}$ and $T$ includes all permutations of $C$ channels, then for sufficiently small $r$:*

$$\frac{N(\mathcal{Z}, \rho_{Z,T}, r)}{N(\mathcal{Z}, \rho_Z, r)} \leq \frac{1}{C!}. \tag{31}$$

*Proof.* For $\rho_Z$-covers under the $\ell_1$-metric, consider the grid:

$$S = \left\{ 2r \cdot \mathbf{k} + r \mid \mathbf{k} \in \{0, 1, \ldots, \lceil 1/(2r) \rceil\}^{n \times C} \right\}. \tag{32}$$

The covering number satisfies $\left(\frac{1}{2r}\right)^{nC} \leq N(\mathcal{Z}, \rho_Z, r) \leq \lceil 1/(2r) \rceil^{nC}$, as the volume of $2r$-cubes covers $[0, 1]^{nC}$.

Define subsets of $S$:

$$S' = \{z \in S \mid \forall 0 \leq i < j < C, z_{0,i} \neq z_{0,j}\},$$
$$S'' = \{z \in S \mid \forall 0 \leq i < j < C, z_{0,i} < z_{0,j}\}.$$

Permuting channels of $S''$ generates $S'$, so $|S'| = C! \cdot |S''|$. The set $(S \setminus S') \cup S''$ forms an $r$-cover for $(\mathcal{Z}, \rho_{Z,T})$.

As $r \to 0^+$, $|S \setminus S'| = O\left((1/r)^{nC-1}\right)$ is negligible compared to $N(\mathcal{Z}, \rho_Z, r) = \Theta\left((1/r)^{nC}\right)$. Thus,

$$\lim_{r \to 0^+} \frac{N(\mathcal{Z}, \rho_{Z,T}, r)}{N(\mathcal{Z}, \rho_Z, r)} = \frac{|S''|}{|S|} = \frac{1}{C!}. \tag{33}$$

$\square$

## C. Probability for Distribution Collision

For continuous distributions, the probability that two independently sampled noise vectors coincide is $0$. However, as neural networks are typically continuous, they may still produce similar outputs for similar noise inputs. We define the coincidence between two noise vectors $\mathbf{z}_1, \mathbf{z}_2$ as $\|\mathbf{z}_1 - \mathbf{z}_2\|_1 \leq \delta$, where $\delta \in \mathbb{R}^+$.

**Proposition C.1.** *Given $n$ vectors $\mathbf{z}_1, \mathbf{z}_2, \ldots, \mathbf{z}_n$ independently sampled from a uniform distribution over $[0,1]^C$, the probability that $\forall i \neq j, \|\mathbf{z}_i - \mathbf{z}_j\|_1 \geq \delta$ is bounded by:*

$$\prod_{i=1}^{n-1} \left[ 1 - i \left( \frac{\delta}{2} \right)^C \right] \quad \text{if } n \leq \left( \frac{\delta}{2} \right)^{-C} + 1, \tag{34}$$

*and $0$ otherwise.*

*Proof.* For each $\mathbf{z}_i$, define a hypercube:

$$D(\mathbf{z}_i) = \left\{ \mathbf{z} \in [0,1]^C \mid \|\mathbf{z} - \mathbf{z}_i\|_1 \leq \frac{\delta}{2} \right\}. \tag{35}$$

To ensure $\|\mathbf{z}_i - \mathbf{z}_j\|_1 \geq \delta$ for all $i \neq j$, the hypercubes $D(\mathbf{z}_i)$ and $D(\mathbf{z}_j)$ must be disjoint. Let $p_m$ denote the probability of placing $m$ non-overlapping hypercubes. The recursion is:

$$p_{m+1} = p_m \left( 1 - m \left( \frac{\delta}{2} \right)^C \right), \tag{36}$$

since each existing hypercube occupies at least $\left( \frac{\delta}{2} \right)^C$ volume in $[0,1]^C$. Solving recursively gives the product bound. For $n > \left( \frac{\delta}{2} \right)^{-C} + 1$, the probability vanishes as the total volume of hypercubes exceeds $1$. $\qquad\square$

When $\delta < 1$, larger $C$ reduces the collision probability due to the exponential decay of $\left( \frac{\delta}{2} \right)^C$.

## D. Proof for Section 4

### D.1. Proof for Theorem 4.4

Since the aggregator produce one single equivariant representation with a set of equivariant representations as input, the transformation must be row-wise on noise matrix.

**Theorem D.1.** *(Invariance) If the aggregator is equivariant to a set of noise transformations $T$, then ENGNN's outputs for graphs and subsets are invariant to $T$.*

*Proof.* Suppose the input noise $Z^{(0)}$ is transformed with row-wise transformation $t$. Let $X_i, Z_i, h_G, h_U$ denote original representations for node $i$, graph, and subset $U$. Let $\hat{X}_i, \hat{Z}_i, \hat{h}_G, \hat{h}_U$ denote the output for transformed input. We are going to prove that $\hat{h}_G = h_G$ and $\hat{h}_U = h_U$.

**Input Layer:** For $k = 0$, the transformed noise is row-wise:

$$\hat{Z}_i^{(0)} = t \left( Z_i^{(0)} \right), \quad \hat{X}_i^{(0)} = X_i^{(0)}. \tag{37}$$

**Equivariance of Message-Passing:** Assume at layer $k$, the node features and noise satisfy:

$$\hat{X}_i^{(k)} = X_i^{(k)}, \quad \hat{Z}_i^{(k)} = t \left( Z_i^{(k)} \right). \tag{38}$$

At layer $k+1$, the update rule preserves equivariance:

$$\left(\hat{X}_i^{(k+1)}, \hat{Z}_i^{(k+1)}\right) = \text{AGGR}_1\Bigg[\left\{\text{AGGR}_2\left(\left\{\left(\hat{X}_j^{(k)}, \hat{Z}_j^{(k)}\right) \mid j \in \mathcal{N}(i)\right\}\right),\right. \tag{39}$$

$$\left.\left(\text{MLP}\left(\hat{X}_i^{(k)}\right), \hat{Z}_i^{(k)}\right)\right\}\Bigg] \tag{40}$$

$$= \text{AGGR}_1\Bigg[\left\{\text{AGGR}_2\left(\left\{\left(X_j^{(k)}, t\left(Z_j^{(k)}\right)\right) \mid j \in \mathcal{N}(i)\right\}\right),\right. \tag{41}$$

$$\left.\left(\text{MLP}\left(X_i^{(k)}\right), t\left(Z_i^{(k)}\right)\right)\right\}\Bigg] \tag{42}$$

$$= \left(X_i^{(k+1)}, t\left(Z_i^{(k+1)}\right)\right). \tag{43}$$

By induction, message-passing layers preserve equivariance.

**Invariance of Graph Representation:**

The graph representation aggregates equivariant node features:

$$\left(\hat{h}_G, \hat{Z}_G\right) = \text{AGGR}\left(\left\{\left(\hat{X}_i, \hat{Z}_i\right) \mid i \in V\right\}\right) \tag{44}$$

$$= \text{AGGR}\left(\{(X_i, t(Z_i)) \mid i \in V\}\right) \tag{45}$$

$$= (h_G, t(Z_G)). \tag{46}$$

Since AGGR is invariant under row-wise transformations, $\hat{h}_G = h_G$.

**Invariance of Subset Representation:**

$$\left(\hat{h}'_U, \hat{Z}'_U\right) = \text{AGGR}_1\left(\left\{\left(\hat{X}_i, \hat{Z}_i\right) \mid i \in U\right\}\right) \tag{47}$$

$$= \text{AGGR}_1\left(\{(X_i, t(Z_i)) \mid i \in U\}\right) \tag{48}$$

$$= (h'_U, t(Z'_U)). \tag{49}$$

**Final Subset Representation:**

$$\left(\hat{h}_U, \hat{Z}_U\right) = \text{AGGR}_2\left[\left\{\left(\text{MLP}\left(\hat{h}_G\right), \hat{Z}_G\right), \left(\hat{h}'_U, \hat{Z}'_U\right)\right\}\right] \tag{50}$$

$$= \text{AGGR}_2\left[\{(\text{MLP}(h_G), t(Z_G)), (h'_U, t(Z'_U))\}\right] \tag{51}$$

$$= (h_U, t(Z_U)). \tag{52}$$

By the equivariance of $\text{AGGR}_2$, $\hat{h}_U = h_U$. $\qquad\square$

### D.2. Proof for Theorem 4.5

**Theorem D.2.** *Fix a graph domain with at most $n_{\max}$ nodes. Assume the noise distribution is continuous and row-exchangeable, and the aggregator in ENGNN achieves universal expressivity on the regular noise space $\mathcal{Z}_{\text{reg}}$ (e.g., the aggregators in Appendix G). Then the expected ENGNN predictor is graph-isomorphism-discriminating for graph-level and subgraph-level tasks on this fixed-size domain.*

*Proof.* **Graph-Level Expressivity.** Consider non-isomorphic graphs $G$ and $H$ in the fixed-size domain, and let $Z \in \mathcal{Z}_{\text{reg}}$ have distinct node features. The universal aggregator can construct injective mappings. Let $u_i, v_i$ denote the invariant and equivariant representations of node $i$, encoding

$$(u_i, v_i) \xrightarrow{\text{inj}} \left(X_i, Z_i, \{\{(X_j, Z_j) \mid (i, j) \in E\}\}\right). \tag{53}$$

Pooling across nodes yields

$$\text{AGGR}\Big(\{\!\{(u_i, v_i) \mid i \in V\}\!\}\Big) \xrightarrow{\text{inj}} \Big(\{\!\{(Z_i, Z_j) \mid (i,j) \in E\}\!\}, \{\!\{(X_i, Z_i) \mid i \in V\}\!\}\Big). \tag{54}$$

Since the $Z_i$ are unique, they act as temporary node identities, so the representation can encode the adjacency matrix and node features up to node relabeling. Therefore non-isomorphic graphs yield distinct representations. Moreover, the same parameterization can map the output set of $G$ over $\mathcal{Z}_{\text{reg}}$ and the output set of $H$ over $\mathcal{Z}_{\text{reg}}$ into disjoint subsets of the output space. The final readout can then map these disjoint subsets to different scalar values, so the expected predictor also separates $G$ and $H$.

As a concrete example, consider two disjoint 3-cycles and one 6-cycle with the same six distinct noise values assigned to corresponding nodes. An MLP can map these six noise values to one-hot identifiers $e_1, \ldots, e_6$. A sum message-passing layer can produce at each node the sum of its own identifier and its neighbors' identifiers. In the 6-cycle, a node can obtain a pattern such as $e_2 + e_3 + e_4$, while this pattern never appears in two disjoint 3-cycles under the corresponding assignment. A pooling layer that counts this pattern separates the two graphs. More generally, unique noise provides temporary node identities, and message passing can encode the edge list.

**Subgraph-Level Expressivity.** For non-isomorphic pairs $(G, U_G)$ and $(H, U_H)$, node representations encode the same information as above:

$$(u_i, v_i) \xrightarrow{\text{inj}} \Big(X_i, Z_i, \{\!\{(X_j, Z_j) \mid (i,j) \in E\}\!\}\Big). \tag{55}$$

Subset aggregation preserves the identities of nodes in the queried subset:

$$\text{AGGR}\Big(\{\!\{(u_i, v_i) \mid i \in U_G\}\!\}\Big) \xrightarrow{\text{inj}} \{\!\{Z_i \mid i \in U_G\}\!\}. \tag{56}$$

The final representation combines subset and full-graph information:

$$\text{ENGNN}(G, U_G, Z) \xrightarrow{\text{inj}} \Big(\{\!\{Z_i \mid i \in U_G\}\!\}, \{\!\{(Z_i, Z_j) \mid (i,j) \in E\}\!\}, \{\!\{(X_i, Z_i) \mid i \in V\}\!\}\Big). \tag{57}$$

Thus non-isomorphic graph–subset pairs have disjoint output sets under regular noise, and the expected subset predictor is isomorphism-discriminating. Because continuous noise lies in $\mathcal{Z}_{\text{reg}}$ almost surely, the expectation over $\mathcal{D}_n$ is well-defined and node-permutation invariant. $\square$

# E. Experimental Setting

Our code is available at https://github.com/MuLabPKU/EquivNoiseGNN. We use PyTorch and PyTorch Geometric for model development. All experiments are conducted on an Nvidia 4090 GPU on a Linux server. We use the AdamW optimizer with a cosine annealing scheduler. We use L1 loss for regression tasks and cross-entropy loss for classification tasks.

We perform random search using Optuna to optimize hyperparameters by maximize valid score. The selected hyperparameters for each model are available in our code. The hyperparameter we tune include number of layer in [2, 10], hidden dimention in [16, 128], number of noise channel in [8, 64], number of noise feature dimension in [16, 64], learning rate in [1e-4, 1e-2], weight decay in [1e-6, 1e-1]. The detailed hyperparameters are in our code. For baselines, we directly use the score reported in the original paper. For ablation model MPNN, NMPNN, we use the same hyperparameter and model architecture as ENGNN, but use aggregators in Appendix G.3. Our experiment on ZINC, ZINC-FULL, and SubgraphCount takes 5 hours per run, and takes less than 1 hour for other tasks per run. All experiments takes about 200 hours.

# F. Dataset

We summarize the statistics of all our datasets in Table 8. For graph datasets, SubgraphCount is the dataset used in substructure counting tasks provided by Huang et al. (2023b), they are random graph with the count of substructure as node label. ZINC, ZINC-FULL (Gómez-Bombarelli et al., 2016), and ogbg-molhiv are three datasets of molecules. LCC, TRI, MDS, and SubgraphCount are all node tasks, but previous GNNs for graph tasks also use them to evaluate expressivity, so we consider them as graph tasks. Ogbg-molhiv is one of Open Graph Benchmark dataset, which aims to use graph structure to predict whether a molecule can inhibits HIV virus replication. For subgraph datasets, we use the code provided

*Table 8.* Statistics of the datasets. #Nodes and #Edges denote the number of nodes and edges per graph. In "Task" column, $k$-CLS means classification with $k$ classes, and REG means regression. In "Split" column, "fixed" means the dataset uses the split provided in the original release, and 10-fold means 10-fold cross validation. Otherwise, it is of the formal training set ratio/valid ratio/test ratio.

| Graph | Name | #Graphs | #Nodes | #Edges | Task | Metric | Split |
|---|---|---|---|---|---|---|---|
| | LCC/TRI/MDS | 3,000 | 20.0 | 30.0 | Node 2-CLS | AUROC | fixed |
| | MUTAG | 188 | 17.9 | 37.6 | 2-CLS | AUROC | 10-fold |
| | NCI1 | 4,110 | 29.9 | 64.6 | 2CLS | AUROC | 10-fold |
| | PROTEINS | 1,113 | 39.1 | 145.6 | 2CLS | AUROC | 10-fold |
| | SubgraphCount | 5,000 | 18.8 | 31.3 | Node REG | MAE | 0.3/0.2/0.5. |
| | ZINC | 12,000 | 23.2 | 24.9 | REG | MAE | fixed |
| | ZINC-full | 249,456 | 23.2 | 24.9 | REG | MAE | fixed |
| | ogbg-molhiv | 41,127 | 25.5 | 27.5 | REG | AUC | fixed |
| Subgraph | Name | #Subgraphs | #Nodes | #Edges | Task | Metric | Split |
| | density | 250 | 5,000 | 29,521 | 3-CLS | F1-score | 0.5/0.25/0.25 |
| | cut-ratio | 250 | 5,000 | 83,969 | 3-CLS | F1-score | 0.5/0.25/0.25 |
| | coreness | 221 | 5,000 | 118,785 | 3-CLS | F1-score | 0.5/0.25/0.25 |
| | ppi_bp | 1,591 | 17,080 | 316,951 | 6-CLS | F1-score | fixed |
| | hpo_metab | 2,400 | 14,587 | 3,238,174 | 6-CLS | F1-score | fixed |
| | hpo_neuro | 4,000 | 14,587 | 3,238,174 | 2-CLS | F1-score | fixed |
| | em_user | 324 | 57,333 | 4,573,417 | 2-CLS | F1-score | fixed |
| Node | Name | - | #Nodes | #Edges | Task | Metric | Split |
| | Cora | | 2,708 | 5,278 | 7-CLS | ACC | 0.6/0.2/0.2 |
| | CiteSeer | | 3,327 | 4,552 | 6-CLS | ACC | 0.6/0.2/0.2 |
| | PubMed | | 19,717 | 44,324 | 3-CLS | ACC | 0.6/0.2/0.2 |
| | Computers | | 13,752 | 245,861 | 10-CLS | ACC | 0.6/0.2/0.2 |
| | Photo | | 7,650 | 119,081 | 8-CLS | ACC | 0.6/0.2/0.2 |
| | Chameleon | | 2,277 | 31,371 | 5-CLS | ACC | 0.6/0.2/0.2 |
| | Squirrel | | 5,201 | 198,353 | 5-CLS | ACC | 0.6/0.2/0.2 |
| | Actor | | 7600 | 26659 | 5-CLS | ACC | 0.6/0.2/0.2 |
| Link | Name | - | #Nodes | #Edges | Task | Metric | Split |
| | Cora | | 2,708 | 5,278 | 2-CLS | Hit@100 | 0.7/0.1/0.2 |
| | Citeseer | | 3,327 | 4,676 | 2-CLS | Hit@100 | 0.7/0.1/0.2 |
| | Pubmed | | 18,717 | 44,327 | 2-CLS | Hit@100 | 0.7/0.1/0.2 |
| | Collab | | 235,868 | 1,285,465 | 2-CLS | Hit@50 | fixed |
| | PPA | | 576,289 | 30,326,273 | 2-CLS | Hit@100 | fixed |
| | DDI | | 4,267 | 1,334,889 | 2-CLS | Hit@20 | fixed |
| | Citation2 | | 2,927,963 | 30,561,187 | 2-CLS | mrr | fixed |

by SubGNN to produce synthetic datasets and use the real-world datasets provided by SubGNN (Alsentzer et al., 2020) directly. For link prediction datasets, random splits use $70\%/10\%/20\%$ edges for training/validation/test set respectively. Different from others, the collab dataset allows using validation edges as input on test set. For node classification datasets, random splits use $60\%/20\%/20\%$ nodes for training/validation/test set respectively.

# G. Equivariant Aggregator

The equivariant aggregator processes a multiset of invariant-equivariant feature pairs $\{(X_i, Z_i) \mid X_i \in \mathbb{R}^d, Z_i \in \mathbb{R}^{L \times C}, i = 1, \ldots, B\}$ to produce an output pair $(X', Z')$. Below, we formalize two variants: $O(C)$(orthogonal transformations on $C$-dimensional noise vectors)- and $S(C)$(permutation on $C$-dimensional noise vectors)-equivariant aggregators, achieving equivariance, universal expressivity, and linear complexity. Prior work (Blum-Smith et al., 2024; Villar et al., 2021; Maron et al., 2020) informs our designs.

## G.1. O(C)-Equivariant Aggregator

The aggregator consists of the following steps:

1. Compute an equivariant orientation matrix $C \in \mathbb{R}^{L' \times C}$ via:

$$C = \sum_{i=1}^{B} f_1(Z_i Z_i^T) Z_i, \quad f_1 : \mathbb{R}^{L \times L} \to \mathbb{R}^{L' \times L}. \tag{58}$$

Projections $C Z_i^T$ yield invariant features.

2. Aggregating Invariants: Aggregate invariant features using a DeepSet (Zaheer et al., 2017) $f_2$:

$$X' = f_2 \left( \{ (X_i, C Z_i^T, Z_i Z_i^T, C C^T) \mid i = 1, \ldots, B \} \right). \tag{59}$$

3. Scale and Aggregating Equivariant Features: Generate equivariant outputs via MLP $f_3 : \mathbb{R}^{d+d+LL'+L'^2} \to \mathbb{R}^L$:

$$Z' = \sum_{i=1}^{B} f_3 \left( X_i, X', C Z_i^T, Z_i Z_i^T, C C^T \right) Z_i. \tag{60}$$

First, we show that this aggregator, denoted as AGGR, is $O(C)$-equivariant.

**Proposition G.1.** *For all $t \in O(C)$, if $\hat{X}', \hat{Z}' = AGGR(\{(X_i, t(Z_i)) | i = 1, \ldots, B\})$ and $X', Z' = AGGR(\{(X_i, Z_i) | i = 1, \ldots, B\})$, then $\hat{X}' = X'$ and $\hat{Z}' = t(Z')$.*

*Proof.* We analyze each step under orthogonal transformation $t$:

1. Orientation: Since $t$ is orthogonal, $t(Z_i Z_i^T) = t Z_i Z_i^T t^T$. The function $f_1$ operates on $Z_i Z_i^T$ and outputs a matrix in $\mathbb{R}^{L' \times L}$. Applying $t$ to $Z_i$ transforms $C$ as:

$$\hat{C} = \sum_{i=1}^{B} f_1(t Z_i Z_i^T t^T) t Z_i = t \left( \sum_{i=1}^{B} f_1(Z_i Z_i^T) Z_i \right) = tC. \tag{61}$$

2. Invariant Aggregation: The input to $f_2$ includes terms like $C Z_i^T$ and $C C^T$. Under transformation, these become:

$$t C Z_i^T t^T, \quad t C C^T t^T. \tag{62}$$

Since $f_2$ is permutation-invariant and operates on multiset inputs, the aggregate $X'$ remains unchanged ($\hat{X}' = X'$).

3. Equivariant Output: The final step applies $t$ to $Z_i$ and computes:

$$\hat{Z}' = \sum_{i=1}^{B} f_3(\cdots) t Z_i = t \left( \sum_{i=1}^{B} f_3(\cdots) Z_i \right) = t Z', \tag{63}$$

where $\cdots = X_i, X', C Z_i^T, Z_i Z_i^T, C C^T$. Thus, the aggregator is $O(C)$-equivariant. $\square$

Second, we show its universal expressivity.

**Proposition G.2.** *There exists a measure zero set $B \subseteq \mathbb{R}^{n \times d} \times \mathbb{R}^{n \times L \times C}$ such that for all $(X, Z), (\hat{X}, \hat{Z}) \in \mathbb{R}^{n \times d} \times \mathbb{R}^{n \times L \times C} \setminus B$, if $\forall P \in S_n, t \in O(C), (PX, Pt(Z)) \neq (\hat{X}, \hat{Z})$, then $AGGR(X, Z) \neq AGGR(\hat{X}, \hat{Z})$.*

*Proof.* Let $B$ exclude inputs where any two $Z_i$ have equal norms ($\|Z_i\|_2 = \|Z_j\|_2$ for $i \neq j$). For $(X, Z) \notin B$, without loss of generality, we assume sorted norms $\|Z_1\| < \|Z_2\| < \cdots < \|Z_B\|$.

Suppose $AGGR(X, Z) = AGGR(\hat{X}, \hat{Z})$. From the invariant aggregation step:

$$\{(X_i, C Z_i^T, Z_i Z_i^T, C C^T)\} = \{(\hat{X}_i, \hat{C} \hat{Z}_i^T, \hat{Z}_i \hat{Z}_i^T, \hat{C} \hat{C}^T)\}. \tag{64}$$

Since norms are distinct and sorted, this implies $X_i = \hat{X}_i$ for all $i$. By Theorem 6.4 in (Blum-Smith et al., 2024), the remaining terms satisfy $C Z_i^T = \hat{C} \hat{Z}_i^T$ and $C C^T = \hat{C} \hat{C}^T$ only if there exists $t \in O(C)$ such that $t Z_i = \hat{Z}_i$. As norms are sorted, $t$ must be the identity, proving expressivity. $\square$

Therefore, there exists a parameterization that the aggregator can differentiate any two different inputs that orthogonal transformations cannot transform them to each other except some rare cases that the equivariant features are very symmetric that the representative orientation degenerates (for example, to zero). These cases are important for other domains using equivariant models, like 3D molecule, where symmetric structures are common and affects molecular properties significantly. However, our equivariant representations are initialized from noise, so the symmetric is very rare.

### G.2. S(C)-Equivariant Aggregator

Following (Maron et al., 2020), we propose permutation equivariant aggregators. This aggregator is permutation-equivariant to node permutations ($S_n$) and noise channel permutations ($S_C$).

The aggregator follows four main steps:

1. **Identifying Noise Channels**: Apply a DeepSet $\psi : \mathbb{R}^{n \times L} \to \mathbb{R}^{L_0}$ to generate unique identifiers for each noise channel:

$$Z^1_{i,:,c} = Z_{i,:,c} \| \psi(Z_{:,:,c}). \tag{65}$$

2. **Node Encoding**: Combine noise and invariant features via a DeepSet $\phi : \mathbb{R}^{(L+L_0) \times C} \to \mathbb{R}^{L_1}$:

$$X^0_i = \phi(Z^1_i) \| X_i. \tag{66}$$

3. **Set Encoding**: Aggregate node features with a DeepSet $\varphi : \mathbb{R}^{k \times (d+L_1)} \to \mathbb{R}^{d_1}$:

$$X^1 = \varphi(X^0). \tag{67}$$

4. **Generating Equivariant Outputs**: Use MLPs $g$ and $h$ to produce outputs:

$$X^2_i = g(X^1 \| X^0_i), \quad Z^2_{i,:,c} = h(X^1 \| X^0_i \| Z^1_{i,:,c}). \tag{68}$$

Each step of the aggregator is efficient, with a time and space complexity of $\Theta(k)$.

Let AGGR denote our aggregator. AGGR guarantees equivariance to node and noise channel permutations. Formally:

**Proposition G.3.** *For any parameterization of $\psi, \phi, \varphi, g, h$, features $X \in \mathbb{R}^{k \times d}$, noise features $Z \in \mathbb{R}^{k \times L \times C}$, and permutations $P_1 \in S_k$ for node, $P_2 \in S_C$ for noise channel, if $X', Z' = AGGR(X, Z)$, then:*

$$P_1(X'), P_2(P_1(Z')) = AGGR(P_1(X), P_2(P_1(Z))). \tag{69}$$

*Proof.* Note that DeepSet model is permutation invariant to permutation on the dimension it aggregates. Moreover, if operator act individually on some dimension, the operator is also equivariant to permutation on the dimension.

Therefore, when $Z \to P_2(P_1(Z))$, $\psi(Z_{:,:,k}) = \psi(Z_{:,:,P_2^{-1}(k)})$, so $Z^1 \to P_2(P_1(Z^1))$.

With $Z^1 \to P_2(P_1(Z^1))$, and $x \to P_1(x)$, $x^0 \to P_1(x^0)$, so $X^1 \to X^1$, $X^2 \to P_1(X^2)$, and $Z^2 \to P_2(P_1(Z^2))$. $\square$

Under mild conditions, AGGR can approximate any equivariant continuous function. Formally:

**Proposition G.4.** *Given a compact set $U \subseteq \mathbb{R}^{k \times d} \times \mathbb{R}^{k \times L \times C}$ that for all each channel of noise has a different elements multiset, AGGR is a universal approximator of continuous $S_k \times S_C$-equivariant functions on $U$.*

*Proof.* Following Segol & Lipman (2020), we first show that the set encoding $X^1$ in our aggregator can approximate any invariant function first. As permutation equivariant function can be expressed as a elementwise transformation conditioned by the invariant function, we can easily approximate any equivariant output.

According to Stone-Weierstrass theorem, to prove the universality of set encoding $X^1$ is equivalent to that our aggregator can differentiate any two input set of invariant and equivariant features with some parameterization.

Let all DeepSet and MLPs in our aggregator be injective. Given two set of features $X \in \mathbb{R}^{k \times d}, Z \in \mathbb{R}^{k \times L \times C}$ and $X' \in \mathbb{R}^{k \times d}, Z' \in \mathbb{R}^{k \times L \times C}$, if $X^1 = X'^1$, then,

- As $\varphi$ is injective, $\exists P_1 \in S_k$, $P_1(X^0) = X'^0$.
- As $P_1(X^0) = X'^0$, $P_1(X) = X'$. Moreover, $P_1(\phi(Z^1)) = \phi(Z'^1)$.
- As $P_1(\phi(Z^1)) = \phi(Z'^1)$, for each row $\phi(Z^1)_i = \phi(Z'^1)_{P_1(i)}$, $\exists P_{2i} \in S_C$, $Z^1_i = P_{2i}(Z'^1_{P_1(i)})$. The permutation of noise channel may be different for each row, but each noise channel is assigned with unique column label, so $P_{2i}$ are all equal to $P_2$. Therefore, $P_2(P_1(Z^1)) = Z'^1$.
- So $P_2(P_1(Z)) = Z'$,

Therefore, the invariant representation is universal. $\square$

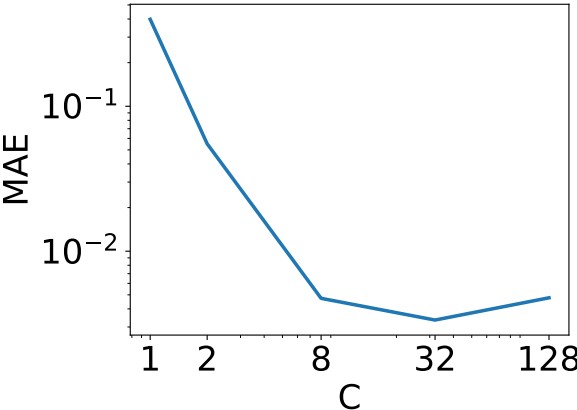

*Figure 2.* Test MAE of triangle counts on subgraph counting task for ENGNN-O with different noise space dimension $C$.

*Table 9.* Roc-auc score $\uparrow$ with uncertainty of ENGNN and rGIN on synthetic.

| dataset | TRI(N) | TRI(X) | LCC(N) | LCC(X) | MDS(N) | MDS(X) |
|---|---|---|---|---|---|---|
| GINs | $0.500_{\pm 0.000}$ | $0.500_{\pm 0.000}$ | $0.500_{\pm 0.000}$ | $0.500_{\pm 0.000}$ | $0.500_{\pm 0.000}$ | $0.500_{\pm 0.000}$ |
| rGINs | $0.908_{\pm 0.013}$ | $0.926_{\pm 0.014}$ | $0.811_{\pm 0.005}$ | $0.852_{\pm 0.007}$ | $0.807_{\pm 0.009}$ | $0.810_{\pm 0.008}$ |
| MPNN | $0.500_{\pm 0.000}$ | $0.500_{\pm 0.000}$ | $0.500_{\pm 0.000}$ | $0.500_{\pm 0.000}$ | $0.500_{\pm 0.000}$ | $0.500_{\pm 0.000}$ |
| NMPNN | $1.000_{\pm 0.000}$ | $1.000_{\pm 0.000}$ | $1.000_{\pm 0.000}$ | $1.000_{\pm 0.000}$ | $0.933_{\pm 0.002}$ | $0.932_{\pm 0.001}$ |
| ENGNN-P | $\mathbf{1.000}_{\pm 0.000}$ | $\mathbf{1.000}_{\pm 0.000}$ | $\mathbf{1.000}_{\pm 0.000}$ | $\mathbf{1.000}_{\pm 0.000}$ | $0.936_{\pm 0.003}$ | $0.934_{\pm 0.004}$ |
| ENGNN-O | $\mathbf{1.000}_{\pm 0.000}$ | $\mathbf{1.000}_{\pm 0.000}$ | $\mathbf{1.000}_{\pm 0.000}$ | $\mathbf{1.000}_{\pm 0.000}$ | $\mathbf{0.938}_{\pm \mathbf{0.006}}$ | $\mathbf{0.939}_{\pm \mathbf{0.002}}$ |

### G.3. Aggregator For Ablation.

For ablation model MPNN, we use aggregator using invariant features only as follows.

1. With MLP $f : \mathbb{R}^d \to \mathbb{R}^{d'}$ and $g' = \mathbb{R}^{d'} \to \mathbb{R}^d$, $X' = g \sum_{i=1}^{n} f(x_i)$.
2. $Z' = 0$.

For ablation model NMPNN, we use aggregator as follows.

1. With MLP $f : \mathbb{R}^{d+CL} \to \mathbb{R}^{d'}$, $C = \sum_{i=1}^{n} f(x_i \| Z_i)$.
2. With MLP $f : \mathbb{R}^{d'} \to \mathbb{R}^d$ and $h : \mathbb{R}^{d'} \to \mathbb{R}^{CL}$, $Z' = h(C)$, and $X' = f(C)$.

## H. Noise Dimension Ablation

We conduct ablation study on triangle counting task. The results are shown in Figure 2. As shown in the Figure, with $C = 1$, the expressivity is low and loss is high. From $C = 1$ to $C = 32$, test loss decrease as $C$ increase, as the expressivity increases. However, from $C = 32$ to $C = 128$, test loss increases, as the noise space gets larger and the sample complexity increases, leading to larger generalization error.

## I. Extra Experiments

We show the uncertainty on synthetic datasets of rGIN (Sato et al., 2021) in Table 9. The standard is small compared with the score gap between baseline rGIN and our ENGNN, validating the effectiveness of our model.

We compare our ENGNN with more noise methods in Table 10.

Following Pellizzoni et al. (2024), we evaluate generalization with limited numbers of training samples and report train–test performance gaps. We use the baselines reported by Pellizzoni et al. (2024). Our ENGNN achieves smaller train–test gaps

*Table 10.* Comparison between our ENGNN and more noise GNN methods.

| Dataset | EXP | CEXP | TRI(N) | TRI(X) | MUTAG | NCI1 | PROTEINS |
|---------|-----|------|--------|--------|-------|------|----------|
| CLIP | 0.99±0.04 | 0.99±0.02 | 0.99±0.00 | 0.81±0.05 | 0.85± 0.09 | 0.81±0.01 | 0.65±0.05 |
| RP | 0.96±0.02 | 0.97±0.02 | 0.99±0.00 | 0.82±0.03 | 0.86± 0.07 | 0.81±0.01 | 0.74±0.04 |
| IRNI | 0.99±0.04 | 0.95±0.14 | 0.99±0.01 | 0.73±0.04 | 0.85±0.05 | 0.82±0.02 | 0.75±0.04 |
| GPSE | 0.74±0.01 | 0.75±0.02 | 0.96± 0.01 | 0.90±0.09 | 0.960±0.019 | 0.826±0.010 | 0.835±0.001 |
| ENGNN-P | **1.00±0.00** | **1.00±0.00** | **1.00±0.00** | **1.00±0.00** | **0.990± 0.019** | **0.897± 0.013** | **0.837± 0.027** |
| ENGNN-O | **1.00±0.00** | **1.00±0.00** | **1.00±0.00** | **1.00±0.00** | **0.990± 0.014** | **0.902± 0.022** | **0.843± 0.028** |

*Table 11.* Train-test performance gap with limited number of training samples on 3-regular graph dataset

| training size | RNI | RP | Tinhofer | ENGNN-O | ENGNN-P |
|---------------|-----|-----|----------|---------|---------|
| 1 | 0.497± 0.001 | 0.502± 0.002 | 0.413± 0.005 | 0.0± 0.0 | 0.0± 0.0 |
| 10 | 0.489± 0.001 | 0.445± 0.003 | 0.213± 0.002 | 0.008± 0.006 | 0.0± 0.0 |
| 100 | 0.310± 0.204 | 0.421± 0.002 | 0.002± 0.0 | -0.007± 0.005 | 0.0± 0.0 |
| 1000 | 0.394± 0.012 | 0.298± 0.059 | 0.0± 0.0 | -0.003± 0.001 | 0.0± 0.0 |

for all training set sizes (Table 11) and better performance with smaller train–test gaps on TU datasets (Table 12).

### I.1. BREC Expressivity Benchmark

We further evaluate ENGNN-O on the BREC expressivity benchmark. We follow the standard routine for random GNNs used by the benchmark: BREC evaluates 100 random node permutations, and in each forward pass ENGNN-O resamples the noise. We use the raw prediction from the linear head in the BREC evaluation routine, without Monte Carlo averaging or thresholding. The model is trained with the BREC training objective. As shown in Table 13, ENGNN-O solves substantially more graph pairs than Graphormer, especially on the Basic and Extension categories. The remaining gap to perfect discrimination reflects the known difference between theoretical universality and finite trained models; prediction variance and numerical stability can cause failures in the benchmark even when the architecture is theoretically expressive.

### I.2. Noise-Handling Ablations

We also test whether the weakness of NMPNN is merely caused by the common resampling protocol. Table 14 compares fixed noise per graph, less frequent resampling, and latent consistency regularization. Across these alternatives, ENGNN-O consistently outperforms NMPNN, indicating that the gain is primarily architectural rather than an artifact of one noise-handling protocol.

*Table 12.* Train-test performance gap on TU datasets. Test mean test AUC. Diff means train-test AUC gap.

| | NCI1 | | MUTAG | | IMDB | | COLLAB | |
|---|---|---|---|---|---|---|---|---|
| | Test | Diff | Test | Diff | Test | Diff | Test | Diff |
| None | 81.8±1.4 | 18.1±1.5 | 81.5±1.3 | 18.4±1.3 | 71.4±3.9 | 2.2±4.4 | 75.3±1.1 | 0.0±1.0 |
| RP | 67.4±1.2 | 32.6±1.2 | 71.2±1.0 | 28.8±1.0 | 63.8±2.3 | 34.6±2.3 | 75.1±2.3 | 18.1±3.1 |
| RNI | 68.0±1.0 | 32.0±1.0 | 72.7±2.4 | 27.3±2.4 | 63.6±8.8 | 36.4±8.8 | 72.1±2.8 | 19.6±4.9 |
| Tinhofer | 72.9±3.0 | 26.9±3.0 | 73.1±2.3 | 26.9±2.3 | 68.6±3.1 | 20.5±3.1 | 75.2±1.7 | 19.5±1.2 |
| Tinhoferw | 81.8±2.3 | 18.1±2.4 | 81.2±1.7 | 18.8±1.7 | 69.4±3.4 | 19.6±3.2 | 80.8±1.9 | 12.2±2.1 |
| LPE | 76.4±1.9 | 23.5±1.9 | 75.4±2.0 | 24.6±2.0 | 68.4±3.3 | 20.7±3.3 | 75.8±1.8 | 19.9±2.3 |
| ENGNN-P | 84.8± 1.5 | 15.1± 1.5 | 91.7± 6.2 | 8.3± 6.2 | 78.7± 1.2 | 4.3± 1.2 | 81.9±1.9 | 1.3± 1.5 |
| ENGNN-O | 85.0± 1.3 | 14.9± 1.3 | 94.4± 5.5 | 5.6± 5.5 | 79.5± 0.5 | 4.1± 0.5 | 81.9± 0.5 | -1.5±0.5 |

*Table 13.* Number of graph pairs distinguished on BREC. Category sizes are shown in parentheses.

| Model | Basic (60) | Regular (140) | Extension (100) | CFI (100) |
|---|---|---|---|---|
| Graphormer | 16 | 12 | 41 | 10 |
| ENGNN-O | **60** | **70** | **98** | **14** |

*Table 14.* Noise-handling ablations on TU datasets (AUROC).

| Method | MUTAG | PROTEINS |
|---|---|---|
| NMPNN | $0.972_{\pm0.054}$ | $0.827_{\pm0.028}$ |
| NMPNN + fixed noise | $0.963_{\pm0.038}$ | $0.787_{\pm0.034}$ |
| NMPNN + resample every 5 epochs | $0.977_{\pm0.031}$ | $0.818_{\pm0.023}$ |
| NMPNN + latent consistency | $0.972_{\pm0.020}$ | $0.813_{\pm0.033}$ |
| ENGNN-O | $\mathbf{0.991_{\pm0.014}}$ | $\mathbf{0.843_{\pm0.028}}$ |
| ENGNN-O + fixed noise | $0.990_{\pm0.060}$ | $0.838_{\pm0.030}$ |
| ENGNN-O + resample every 5 epochs | $0.977_{\pm0.032}$ | $0.838_{\pm0.037}$ |
| ENGNN-O + latent consistency | $0.983_{\pm0.028}$ | $0.841_{\pm0.033}$ |

