# OpenReview forum: "Expressive Graph Neural Networks via Equivariant Use of Noise"
_ICML.cc/2026/Conference — ICML 2026 spotlight_

### Official Review · Reviewer_Bwx2 · 2026-03-02

**Soundness:** 4
**Presentation:** 3
**Significance:** 4
**Originality:** 4
**Overall Recommendation:** 5
**Confidence:** 3

**Summary:**

The authors solve a gap between GNNs which lack in expressivity, and theoretical universal-expressive models which show poor generalisation.

They solve this issue by applying random noise to nodes in an equivariant manner that preserves symmetry while allowing non-isomorphism detection.

This is a strong concept paper with results showing that their ENGNN model approaches universal expressivity and high performance.

They outperform standard MPNNs and naive noise MPNNs, though time and memory costs are drastically increased.

Benchmark results show best or close to best performance on most tasks.

The main drawbacks are increased computational overhead due to the addition of noise features, and the method costs extra work to be integrated with existing models.

**Compliance With Llm Reviewing Policy:**

Affirmed.

**Key Questions For Authors:**

1. " on molhiv, ENGNN is outperformed by GNNAK+ and I2GNN, which may be attributed to the inductive bias provided by subgraph-based GNNs. " - Can you explain this a little more?

**Limitations:**

The future works section is pretty minor (probably a page limit issue), how are you planning to reduce computation costs?

**Strengths And Weaknesses:**

**Soundness**
- Theory and proofs are well stated and give a good mathematical explanation of their methods.
- The benchmark successes support the methods stated strengths.

**Presentation**
- Figure 1 does a good job of expressing the papers focus, while the tables are clear and informative and follow standards that make it easy to immediately understand.
- The appendix is quite large, and particularly B is a slog for me to read through, though there isn't really an issue with that as the information is necessary.

**Significance**
- This method could become a new standard beyond MPNNs.
- However, the increase in memory and time usage worries me, particularly with the claim that this will improve drug discovery field, in which models tend to already take vast amounts of time and hit OOM due to the massive datasets.
- And it does not outperform the competition on all benchmarks.

**Originality**
- I quite like the use of equivariance on the noise transformations, this is novel and very interesting.
- Having the invariant and equivariant representations separate is a great way to handle the concept

---

> ### Author Rebuttal · Authors · 2026-03-31
>
> We thank the reviewer for the insightful feedback. We address your concerns and questions below:
>
> * Q1: " on molhiv, ENGNN is outperformed by GNNAK+ and I2GNN, which may be attributed to the inductive bias provided by subgraph-based GNNs. " - Can you explain this a little more?
>
> As shown in [1], molecular graphs typically exhibit a hierarchical structure (e.g., atoms $\rightarrow$ functional groups $\rightarrow$ whole molecule). Subgraph-based GNNs explicitly decompose the global graph into smaller subgraphs, providing a strong inductive bias that allows them to capture these functional group substructures more effectively than general-purpose architectures.
>
> * L1: The future works section is pretty minor (probably a page limit issue), how are you planning to reduce computation costs?
>
> The primary computational overhead of ENGNN stems from the simultaneous processing of both invariant and equivariant feature streams, whereas standard GNNs only process a single feature type. To mitigate this, we plan to explore a hybrid architecture where the noise augmentation and equivariant processing are restricted to a subset of layers (like a few first layers). By utilizing standard GNN layers for the remainder of the network, we can significantly reduce the overall parameter count and FLOPs while still benefiting from the enhanced expressivity provided by the equivariant channels in key stages of the model.
>
>
> [1] An Efficient Subgraph GNN with Provable Substructure Counting Power. KDD 2024.

---

> > ### Author Rebuttal · Reviewer_Bwx2 · 2026-04-02
> >
> > My questions are resolved, I thank the authors for sharing of the subgraphing paper.

---

### Official Review · Reviewer_RDSu · 2026-03-05

**Soundness:** 2
**Presentation:** 3
**Significance:** 3
**Originality:** 3
**Overall Recommendation:** 4
**Confidence:** 3

**Summary:**

This paper proposes ENGNN: instead of injecting i.i.d. random node noise, the model enforces equivariance/invariance to a group of transformations applied to each node’s noise. The paper claims improved generalization via reduced sample complexity, universal expressivity under suitable aggregators, and strong empirical results across graph/node/link/subgraph tasks.

**Compliance With Llm Reviewing Policy:**

Affirmed.

**Final Justification:**

The provided 2-round response is helpful; I would like to raise the score to 4.

**Key Questions For Authors:**

1. How sensitive are results to noise dimension $C$, group choice (orthogonal vs permutation), and whether noise is fixed vs resampled?

2. Add the definition of the equivariant aggregator and better transitions in the main text.

3. Please clarify weaknesses 2 and 4.

4. The appendix contains some technical issues that hurt the paper. E.g., the graph semi-metric seems incorrect for permutation-invariant graph comparison: as in lines 706-708, $\rho_G(G_1,G_2)=\lVert X_1-X_2\rVert+\lVert A_1-A_2\rVert$ is defined, and as in line 708, $S_n$ (the symmetric group) is mentioned but no minimization over permutations is used, so $\rho_G$ depends on node ordering. The Lemma B.4 uses $\ln N(\mathcal H,\rho_H,\cdot)$ (lines 718-722), yet as in lines 785-788 (Eq.28) the bound multiplies $N(\mathcal G,\cdot)$ and $N(\mathcal Z,\cdot)$ rather than placing them inside logs, which looks inconsistent with the preceding derivation. The expressivity proof remains somewhat hand-wavy and leans on strong injectivity/universality assumptions about the aggregator (as in lines 935-948). That said, the quality of the appendix in its current form is mixed.

If the questions are properly resolved during rebuttal, the reviewer will raise the score.

**Limitations:**

yes.

**Strengths And Weaknesses:**

Strengths

1. Simple and clear idea: treat noise as a symmetry-respecting signal rather than an arbitrary node-ID surrogate. This directly targets the known “noise helps expressivity but breaks generalization” issue (Fig. 1).

2. Theory direction is reasonable: framing invariance as shrinking effective hypothesis complexity via a semi-metric / covering-number argument is conceptually aligned with generalization under invariances.

3. Board empirical experiments: including whole-graph, node, link, subgraph with competitive results, and some evidence of efficiency.

Weaknesses

1. The empirical claim seems overstated: "Our ENGNN achieves best or second best performance on 5/6 datasets." as in lines 364-365. But, in Table 5, the PPA results for ENGNN are not the best/second among listed methods (as in line 401-402, ENGNN-P=44.97 and ENGNN-O=48.44 on PPA, below Neo-GNN=49.13 (line 396) and SEAL=48.80 (line 394), and far below BUDDY=89.85 (line 397)).

2. Sample complexity expression confusion: as in lines 181-183, Eq. (3) scales with $N_{Z, T}N_G \ln(N_Y)$. Typical uniform convergence bounds depend on logs of covering numbers, so this looks suspicious unless $N_{Z, T}/N_G $ are already log-covering terms.

3. Theory implementation gap in main text: the equivariant aggregator is deferred to Appendix G, while the main body equations are hard to parse for how invariance is guaranteed after mixing streams.

4. Noise/theory: as in line 304, “the noise is resampled in each forward pass". It’s unclear whether the generalization analysis assumes fixed noise per graph, per epoch, or per forward pass? And whether invariance is enforced pointwise or in expectation.

5. Complexity claims need conditions: as in lines 226-229, the paper notes universal expressivity may require large depth/width (and potentially non-linear cost), while as in lines 272-273, it states $O(n+m)$ complexity.

---

> ### Author Rebuttal · Authors · 2026-03-31
>
> We thank the reviewer for the thoughtful and detailed feedback. We address your concerns and questions below:
>
> * W1: The empirical claim seems overstated: "Our ENGNN achieves best or second best performance on 5/6 datasets." as in lines 364-365. But, in Table 5, the PPA results for ENGNN are not the best/second among listed methods.
>
> There may exist some misunderstanding. There are 6 datasets in total. Our claim was intended to highlight that ENGNN achieves 1st or 2nd place on all datasets except PPA. We will revise the claim in the final version to: "Our ENGNN achieves best or second-best performance on 5 out of 6 evaluated datasets."
>
> * W2&Q4.2: Sample complexity expression confusion: as in lines 181-183, Eq. (3) scales with $N_{Z,T}N_G\ln N_{Y}$. Typical uniform convergence bounds depend on logs of covering numbers, so this looks suspicious unless $N_{Z,T}/N_G$ are already log-covering terms.
>
> The $N_{Z,T}/N_G$ is the covering number of input space, and we insist that our conclusion is correct. This derivation follows the established methodology in theorem 18 of Section 5.2 of [1], which also moves covering number of input space outside log.
>
> The problem roots in the difference between the hypothesis function space and input data space. While uniform convergence bounds typically feature log of covering number of hypothesis space $\ln N(\mathcal{H})$ as shown in our Lemma B.4.
> In Lemma B.5, we prove the covering number of the hypothesis space $N(\mathcal{H})$ is proportional to $N_Y^{N_{Z,T}N_G}$. Consequently, when taking the logarithm ($\ln N_Y^{N_{Z,T}N_G}$), the input space covering number $N_{Z,T}$ and $N_G$ move outside the log.
>
> * W3&Q4.2: Theory implementation gap in main text: the equivariant aggregator is deferred to Appendix G. The expressivity proof remains somewhat hand-wavy and leans on strong injectivity/universality assumptions about the aggregator (as in lines 935-948).
>
> We postponed the specific design and universality proof of equivariant aggregators to Appendix G because these components build upon established geometric GNN literature [2, 3] rather than being a standalone architectural novelty of this paper. Our contribution lies in being the first to leverage these equivariant aggregators for general graph learning via noise augmentation.  However, we agree that visibility is key. We will move the formal definition of the equivariant aggregator (mapping pairs $\{(X^{(k)}_i, Z^{(k)}_i )\}$ to $(X', Z')$) and provide a smoother transition in the main text.
>
> * W4. Noise/theory: as in line 304, “the noise is resampled in each forward pass". It’s unclear whether the generalization analysis assumes fixed noise per graph, per epoch, or per forward pass? And whether invariance is enforced pointwise or in expectation.
>
> In our generation analysis, we consider noise as a part of feature, and the input graph-noise pair distribution is composed two independent graph and noise distribution, so the generalization analysis assumes fixed noise per forward pass.
>
> The invariance is enforced pointwise. As shown in Appendix D.1, given noise transformer $t$ to which aggregator is equivariant, the model output is invariant: $ENGNN(G, Z)=ENGNN(G, t(Z))$ for all input graph $G$ and input noise $Z$, .
>
>
> * W5. Complexity claims need conditions: as in lines 226-229, the paper notes universal expressivity may require large depth/width (and potentially non-linear cost), while as in lines 272-273, it states $O(N+M)$ complexity.
>
> We will clarify the complexity is under the setting that depth and width are fixed.
>
> * Q1. How sensitive are results to noise dimension $C$, group choice (orthogonal vs permutation), and whether noise is fixed vs resampled?
>
> As shown in Appendix H, performance is not sensitive to $C\ge 16$; dimensions that are too small can harm expressivity.
>
> All our experiments in section 5 provides performance for both ENGNN-P (permutation) and ENGNN-O (orthogonal). In general ENGNN-O performs better, but they both achieves outstanding performance.
>
> Noise is resampled in each forward pass to maintain consistency with existing noise-augmented GNN baselines.
>
>
> * Q4.1 The appendix contains some technical issues that hurt the paper. E.g., the graph semi-metric seems incorrect for permutation-invariant graph comparison
>
> Our primary focus is analyzing the symmetry of the noise channel rather than node-order symmetry. We used a simplified semi-metric; however, our proof is not dependent on the specific form of $\rho_G$. We can replace it with the permutation-invariant version: $\rho(G_1, G_2)=\min_{P\in S_n}||PX_1-X_2||^2+||PA_1P^T-A_2||^2$. This remains a semi-metric and does not change the validity of our proof and conclusions.
>
> [1] Distance-Based Classification with Lipschitz Functions. JMLR 2004.
>
> [2] On learning sets of symmetric elements. ICML 2020.
>
> [3] Scalars are universal: Equivariant machine learning, structured like classical physics. NeurIPS 2021.

---

> > ### Author Rebuttal · Reviewer_RDSu · 2026-03-31
> >
> > Thank you for the detailed rebuttal. The clarification of the empirical claim, the statement that the generalization analysis assumes fixed noise per forward pass, the clarification that invariance is enforced pointwise, and the note that the complexity claim is under fixed depth/width are all helpful. However, I remain only partially convinced.
> >
> > 1. While the rebuttal explains the sample-complexity expression by distinguishing hypothesis-space and input-space covering numbers, the derivation is still not fully transparent to me.
> >
> > 2. Some technical concerns still feel only partially addressed, e.g., the semi-metric / permutation issue and the reliance of the expressivity proof on strong injectivity/universality assumptions about the aggregator.

---

> > > ### Author Response · Authors · 2026-04-07
> > >
> > > We are glad to hear that our reply partially addressed your previous concerns. We now address your remaining questions below. We apologize that we were unable to complete our reply before your final assessment. If you find that our response resolves your concerns, we would greatly appreciate it if you could reconsider your score, as it is important for a fair evaluation of our work.
> > >
> > > AQ1. While the rebuttal explains the sample-complexity expression by distinguishing hypothesis-space and input-space covering numbers, the derivation is still not fully transparent to me.
> > >
> > > Our derivation follows the established methodology in Theorem 18 of Section 5.2 of [1]. The conversion from the covering number of the hypothesis space to that of the data space is primarily carried out in our Lemma B.5, which, in brief, shows that $N_H\le N_Y^{N_GN_Z}$, where $N_H$, $N_Y$, $N_G$, and $N_Z$ are the covering numbers of the hypothesis space, the output y space, the graph space
> > > G, and the noise space Z, respectively.
> > >
> > > The proof sketch is as follows. Since the hypothesis function (mapping a (graph, noise) pair to an output y) is Lipschitz, its behavior can be approximated by its values on a finite set of input sample points (i.e., (graph, noise) pairs). Specifically, we select $N_G$ representative graphs in the graph space, $N_Z$ points in the noise space, and $N_Y$ points in the output space. **The number of possible input pairs is $N_GN_Z$, and each input pair can be mapped to one of $N_Y$ possible outputs, so the number of possible discretized functions is $N_Y^{N_GN_Z}$.** These discretized functions suffice to cover the entire function space, yielding $N_H\le N_Y^{N_GN_Z}$
> > >
> > > AQ2.1. Some technical concerns still feel only partially addressed, e.g., the semi-metric / permutation issue.
> > >
> > > We appreciate the reviewer's careful attention to this point and offer the following clarifications:
> > >
> > > 1. Our goal is to prove generalization for general graph models, and our focus is not the node order invariance. Accordingly, we adopt $\rho_G=\Vert A_1-A_2\Vert^2+\Vert X_1-X_2\Vert^2$, which is sensitive to node order. This choice makes the result also valid to models that are affected by node order, such as methods based on graph canonicalization [6].
> > >
> > > 2. For graph models that are invariant to node permutation, our current bound remains valid in this case: It neglects models' invariance to node order permutation and thus consider hypothesis space with both invariant models and models not invariant, leading to larger hypothesis space and thus larger sample complexity upper bound. The bound is still valid and is simply less tight.
> > >
> > > 3. Our proof does not rely on any specific property unique to the graph semi-metric. Therefore, if the semi-metric is replaced with a different (semi-)metric, the proof and its conclusions remain valid. For node-order invariant model, one could substitute a permutation-invariant semi-metric, which would yield a tighter bound.
> > >
> > > AQ2.2. The reliance of the expressivity proof on strong injectivity/universality assumptions about the aggregator.
> > >
> > > The strong expressivity of the aggregator is a foundational requirement for establishing the expressivity of GNNs, dating back to GIN [2]. Prior work on noise-augmented GNNs [3] similarly requires strong aggregator expressivity to achieve universal approximation. Therefore, **the dependence of our expressivity proof on the injectivity/universality of the aggregator is standard practice in GNN expressivity analysis.**
> > >
> > > Moreover, the universality of the aggregators we employ has been rigorously established in the context of geometric GNNs [4, 5]. We also provide a self-contained proof of these aggregators' universal expressivity in our Appendix G.
> > >
> > >
> > > [1] Distance-Based Classification with Lipschitz Functions. JMLR 2004.
> > >
> > > [2] How Powerful Are Graph Neural Networks? ICLR 2019.
> > >
> > > [3] The Surprising Power of Graph Neural Networks with Random Node Initialization. IJCAI 2021.
> > >
> > > [4] On Learning Sets of Symmetric Elements. ICML 2020.
> > >
> > > [5] Scalars Are Universal: Equivariant Machine Learning, Structured Like Classical Physics. NeurIPS 2021.
> > >
> > > [6] Rethinking the Power of Graph Canonization in Graph Representation Learning with Stability. ICLR 2024.

---

### Official Review · Reviewer_zXBP · 2026-03-13

**Soundness:** 3
**Presentation:** 3
**Significance:** 2
**Originality:** 2
**Overall Recommendation:** 5
**Confidence:** 3

**Summary:**

### **Core Problem**

- **Expressivity vs. Generalization Gap:** Standard Message Passing Neural Networks (MPNNs) face fundamental expressivity limitations on complex tasks. While theoretically universal models (like high-order GNNs) exist, they suffer from high computational costs or poor generalization, limiting real-world use.

- **The Flaw of Naive Noise:** Augmenting nodes with random noise features is a task-agnostic way to achieve universal expressivity theoretically. However, naively injecting noise dramatically increases the model's input space and breaks graph symmetries, leading to poor generalization in practice.


### **Proposed Method: Equivariant Noise GNN (ENGNN)**

- **Equivariance to Noise Transformations:** To solve the generalization issue, ENGNN uses random noise features but enforces equivariance to nodewise noise transformations, such as orthogonal transformations or channel permutations.

- **Two-Stream Architecture:** The model uses a dual-stream design. An invariant stream is initialized with standard node features, while an equivariant stream is initialized with random noise. A specially designed aggregator layer mixes these streams while preserving their symmetry properties.


### **Theoretical Contributions**

- **Generalization:** The authors establish a sample complexity bound from PAC learning theory. They mathematically prove that enforcing invariance to chosen noise transformations reduces theoretical sample complexity, which yields a tighter generalization bound compared to naive noise injection.

- **Universal Expressivity:** The paper proves that when equipped with a universally expressive aggregator, ENGNNs achieve theoretical universal expressivity across graph-level, node-level, link-level, and subgraph-level prediction tasks.


### **Empirical Results**

- **Practical Scalability:** The authors implement two variants: ENGNN-O (equivariant to orthogonal transformations) and ENGNN-P (equivariant to permutation). In practice, these models maintain the linear time and space complexity of standard MPNNs.

- **Performance:** Extensive experiments across node, link, subgraph, and graph-level tasks demonstrate that ENGNNs consistently outperform vanilla MPNNs and naive Noise MPNNs. They also achieve performance comparable to highly expressive, computationally expensive specialist models.

**Compliance With Llm Reviewing Policy:**

Affirmed.

**Final Justification:**

Thank you for the additional clarifications. This follow-up addresses my remaining concerns more satisfactorily.

(Overall Recommendation: 4 -> 5)

For [Q1], the most helpful point is the explicit statement that the advantage of ENGNN over RP is not a different expressivity class, but rather practical efficiency, scalability, and better generalization while retaining the same universal-expressivity ceiling. This directly clarifies the contribution calibration I was asking about. In particular, it makes clear that the main contribution should be interpreted primarily as a practically stronger architectural formulation rather than a strictly stronger expressivity result.

For [Q2], the additional ablations with fixed noise, less frequent resampling, and latent consistency are useful. These results make a stronger case that ENGNN’s gains are primarily architectural rather than an artifact of a particular noise-handling protocol. I found this new evidence materially helpful.

Overall, my concerns are now adequately addressed. I still view the main contribution as stronger on the practical efficiency/generalization side than on introducing a new expressivity tier, but the authors have now made that distinction clearly.

**Key Questions For Authors:**

- **\[Q1]** While the authors claim ENGNN is unique in applying equivariance to the noise space to improve a general GNN, the conceptual novelty threshold needs stricter delineation. Relational Pooling (RP) already tackles the symmetry of augmented random IDs by marginalizing over the permutation group to maintain invariance. While ENGNN processes continuous noise channels through an equivariant aggregator rather than averaging discrete ID permutations, the fundamental philosophy of handling augmented random features via symmetry groups is highly similar. Can you more rigorously delineate the theoretical novelty of ENGNN-P's permutation equivariance over noise channels compared to the permutation invariance achieved in Relational Pooling? Does your method unlock a different tier of expressivity, or is the primary contribution purely computational efficiency?

- **\[Q2]** The authors note that the naive noise baseline (NMPNN) resamples noise during every forward pass, following the protocol established by Abboud et al. While ENGNN clearly outperforms this setup, a rigorous evaluation must ask if the poor generalization of NMPNN is fundamentally an architectural flaw, or if it is artificially exacerbated by this specific, highly volatile training dynamic. Could the downstream performance degradation of naive noise methods be partially mitigated by alternative noise regularization techniques (e.g., consistency losses across different noise samples) rather than strictly requiring architectural equivariance?

**Limitations:**

Yes. The authors have explicitly addressed both in the manuscript:

- Limitations: In Section 7, they acknowledge that introducing noise features creates additional computational overhead. They also note that their method requires modifications to the aggregator, meaning it is not a simple "plug-and-play" module for existing GNNs.

- Societal Impact: They include an "Impact Statement" outlining potential positive downstream applications, such as accelerating drug discovery and improving traffic efficiency. They explicitly state that they do not foresee specific potential risks that need highlighting.

**Strengths And Weaknesses:**

# Strengths

- **\[S1] Theoretical Grounding of the Noise-Generalization Trade-off.** The authors provide a rigorous and highly convincing mathematical formalization of why naive noise fails. By utilizing PAC learning theory, they prove that for a naive Noise MPNN, the covering number of the noise space grows exponentially (up to $2^{nC}$). This sample complexity analysis provides a robust theoretical foundation for the intuitive problem that breaking graph symmetries with random features inflates the hypothesis space and ruins generalization.

- **\[S2] Empirical Validation of Downstream Harm.** The paper provides definitive empirical evidence that unconstrained noise harms downstream performance through well-designed ablation studies. The comparison between the standard MPNN, the naive Noise MPNN (NMPNN), and ENGNN across multiple tasks perfectly isolates the architectural benefit of equivariance.

- **\[S3] Clear Positioning and Literature Context.** The authors demonstrate strong scholarship by transparently acknowledging that they are not the first to spot the naive noise problem, accurately citing prior mitigation attempts like noise resampling. They successfully justify why their method is superior to existing non-naive methods. They accurately frame methods like MPLP or GPSE as models that collapse noise into fixed heuristics (like Laplacian eigenvectors), which solves the generalization issue but sacrifices the theoretical universal expressivity that makes noise appealing.


# Weaknesses

- See \[Key Questions For Authors]

---

> ### Author Rebuttal · Authors · 2026-03-31
>
> We thank the reviewer for the insightful feedback. We address your concerns and questions below:
>
> 1. [Q1] While the authors claim ENGNN is unique in applying equivariance to the noise space to improve a general GNN, the conceptual novelty threshold needs stricter delineation. Relational Pooling (RP) already tackles the symmetry of augmented random IDs by marginalizing over the permutation group to maintain invariance. While ENGNN processes continuous noise channels through an equivariant aggregator rather than averaging discrete ID permutations, the fundamental philosophy of handling augmented random features via symmetry groups is highly similar. Can you more rigorously delineate the theoretical novelty of ENGNN-P's permutation equivariance over noise channels compared to the permutation invariance achieved in Relational Pooling? Does your method unlock a different tier of expressivity, or is the primary contribution purely computational efficiency?
>
> We appreciate the opportunity to clarify the distinction between our approach and Relational Pooling (RP). While both methods address symmetry in the context of noise/ID augmentation, they operate on different dimensions of the noise space and utilize fundamentally different mechanisms:Dimensionality of Symmetry: Let the input noise be a matrix $Z \in \mathbb{R}^{N \times C}$. Relational Pooling focuses on the symmetry of node permutations (the first dimension, $N$). It achieves invariance by sampling or marginalizing over various node orderings. In contrast, ENGNN focuses on the symmetry of noise channels (the second dimension, $C$).Mechanism: RP relies on sampling discrete permutations to maintain invariance. ENGNN achieves equivariance through a specialized equivariant GNN architecture designed to process continuous noise channels.By enforcing equivariance directly within the architecture rather than through sampling, ENGNN ensures a consistent representation of the noise space that is not subject to sampling variance. We will clarify in the revision that ENGNN is unique in applying an equivariant architecture to the noise channels to improve general GNN expressivity.
>
>
> 2. [Q2] The authors note that the naive noise baseline (NMPNN) resamples noise during every forward pass, following the protocol established by Abboud et al. While ENGNN clearly outperforms this setup, a rigorous evaluation must ask if the poor generalization of NMPNN is fundamentally an architectural flaw, or if it is artificially exacerbated by this specific, highly volatile training dynamic. Could the downstream performance degradation of naive noise methods be partially mitigated by alternative noise regularization techniques (e.g., consistency losses across different noise samples) rather than strictly requiring architectural equivariance?
>
>
> We find that consistency losses on prediction across different noise samples is not very effective on TU dataset. The reason may be that the supervised learning loss already trains prediction with different noise input to the same target. On TU datasets, we use l2 loss to align the prediction of the same graph with different input noise. The results are as follows,
>
> |Method |MUTAG | PROTEINS|
> |-|-|-|
> | NMPNN | 0.972±0.054|0.827±0.028|
> |NMPNN+consistency loss| 0.968±0.012|0.819±0.024|
> |ENGNN-O|0.991±0.014|0.843±0.028|

---

> > ### Author Rebuttal · Reviewer_zXBP · 2026-04-01
> >
> > Thank you for the rebuttal. The additional clarification was helpful, and I found the new response to [Q2] more convincing than the response to [Q1].
> >
> > For [Q1], I appreciate the clearer distinction from Relational Pooling. The explanation that RP addresses symmetry over node permutations whereas ENGNN focuses on noise-channel transformations, and that ENGNN enforces this through an equivariant architecture rather than sampling/marginalization, does clarify an important implementation-level difference. However, my original concern was mainly about the conceptual novelty threshold. On that point, I remain only partially convinced. The rebuttal helps me understand the intended framing better, but it does not fully establish whether ENGNN provides a genuinely different expressivity regime relative to RP-style approaches, or whether the main advance is better computational efficiency / reduced sampling variance / better practical generalization. In other words, this part feels better framed, but not fully resolved at the level of originality.
> >
> > For [Q2], the added consistency-loss experiment is useful. It is relevant to my concern, and the reported results suggest that simply regularizing NMPNN across noise samples does not recover the gap to ENGNN, at least on the reported TU datasets. This strengthens the case that the benefit is not merely an artifact of the specific resampling protocol. At the same time, I still view this as somewhat limited evidence, since it is a narrow intervention rather than a broader sweep over alternative stabilization strategies. Still, this point is more substantially addressed than [Q1].
> >
> > Overall, the rebuttal improves my understanding of the paper and provides meaningful additional evidence on [Q2]. My remaining reservation is primarily about how to calibrate the contribution: specifically, whether the paper should be read as introducing a fundamentally new symmetry-based idea, or as a practically strong and technically sound architectural instantiation of a closely related existing philosophy. That distinction matters for my final view of originality and significance.
> >
> > ## Follow-up questions
> > 1. For **[Q1]**, could you state more explicitly whether your claimed advantage over RP is primarily:
> >
> >    * a different expressivity class,
> >    * better computational scalability / lower variance,
> >    * or better practical generalization under noise augmentation?
> >
> > 2. Can you identify any setting in which ENGNN-P is provably or empirically stronger than RP in a way that is **not** reducible to sampling efficiency or variance reduction?
> >
> > 3. For **[Q2]**, did you try any alternatives beyond the reported consistency loss, such as fixing one noise draw per graph, less frequent resampling, or stronger invariance regularization? Even a brief negative-result summary would help separate architectural from optimization effects.

---

> > > ### Author Response · Authors · 2026-04-07
> > >
> > > We are glad to hear that our previous reply address some of your concerns.
> > >
> > > AQ1. For [Q1], could you state more explicitly whether your claimed advantage over RP is primarily:
> > > 1. a different expressivity class,
> > > 2. better computational scalability / lower variance,
> > > 3. or better practical generalization under noise augmentation?
> > > 4. Can you identify any setting in which ENGNN-P is provably or empirically stronger than RP in a way that is not reducible to sampling efficiency or variance reduction?
> > >
> > > Our primary contribution lies in making noise-augmented GNNs practically usable. ENGNN achieves better practical generalization (as demonstrated in Section 5 of our paper, where ENGNN attains strong performance across various real-world benchmarks) with practical scalability (requiring only a single network forward pass, rather than averaging over node orderings as in RP, leading to cost comparable to vanilla MPNNs, as shown in our Table 7).
> > >
> > > Regarding theoretical expressivity, Relational Pooling has already achieved universal expressivity—the maximal expressivity level (as shown in Theorem 2.1 of the RP paper [1])—and no strictly stronger expressivity class is defined in the current graph learning setting. Our ENGNN also achieves universal expressivity. Therefore, the advantage of ENGNN over RP is not a different expressivity class, but rather its practical efficiency and generalization: ENGNN attains the same theoretical ceiling with better real-world performance.
> > >
> > > AQ2. For [Q2], did you try any alternatives beyond the reported consistency loss, such as fixing one noise draw per graph, less frequent resampling, or stronger invariance regularization? Even a brief negative-result summary would help separate architectural from optimization effects.
> > >
> > > We thank the reviewer for this constructive suggestion. The results are summarized below:
> > >
> > > |Method|	MUTAG|	PROTEINS|
> > > |-|-|-|
> > > |NMPNN|	0.972±0.054|	0.827±0.028|
> > > |+ fixing one noise|0.963±0.038	|0.787±0.034	|
> > > |+ change noise every 5 epoches| 0.977±0.031  |0.818±0.023	|
> > > |+ latent consistency| 0.972±0.020 |0.813±0.033	|
> > > |ENGNN-O|	0.991±0.014|	0.843±0.028|
> > > |+ fixing one noise|  0.990± 0.060|	0.838±0.030|
> > > |+ change noise every 5 epoches|  0.977± 0.032 | 0.838± 0.037	|
> > > |+ latent consistency| 0.983± 0.028 | 0.841±0.033	|
> > >
> > > Several observations can be drawn:
> > >
> > > 1. Fixing one noise draw per graph hurts NMPNN noticeably (PROTEINS drops from 0.827 to 0.787), as the model overfits to a single noise realization and loses the regularization benefit of resampling. For ENGNN-O, accuracy is largely preserved but variance increases substantially (MUTAG: 0.014 → 0.060), suggesting that ENGNN's equivariant architecture is more robust to this change, though resampling still helps stabilize training.
> > >
> > > 2. Less frequent resampling (every 5 epochs) yields mixed results: a slight improvement on MUTAG for NMPNN but a decline on PROTEINS, and a consistent slight decline for ENGNN-O. This indicates that frequent resampling provides a useful data-augmentation effect that benefits both architectures.
> > >
> > > 3. Latent consistency regularization: Consistency loss on latent produced by last layer rather than on prediction. It effectively reduces variance for NMPNN (MUTAG std: 0.054 → 0.020) without harming mean accuracy, confirming its role as a stabilizer. For ENGNN-O, the effect is more modest, as the equivariant architecture already provides inherent stability.
> > >
> > > 4. Across all noise-handling strategies, ENGNN-O consistently outperforms NMPNN, demonstrating that the performance gain of ENGNN is primarily architectural rather than an artifact of a particular optimization strategy.
> > >
> > > [1] Relational Pooling for Graph Representation. ICML 2019.

---

### Official Review · Reviewer_L1SF · 2026-03-13

**Soundness:** 1
**Presentation:** 2
**Significance:** 2
**Originality:** 3
**Overall Recommendation:** 4
**Confidence:** 3

**Summary:**

The authors propose Equivariant Noise GNNs (ENGNNs), a framework that uses random noise features to boost GNN expressivity (by introducing new aggregators). The key insight is that enforcing symmetry in the noise space reduces the covering number, improving sample complexity and thus generalization compared to naive noise injection. They argue that their approach is universal approximator, provide PAC-based bounds, and evaluate their model across node, link, subgraph, and graph-level tasks.

**Compliance With Llm Reviewing Policy:**

Affirmed.

**Final Justification:**

Overall, the authors’ rebuttal has alleviated some of my concerns, particularly regarding the validity of the expressivity results (e.g., clarifying permutation invariance in expectation) and the inclusion of additional experiments. In light of this, I have increased my score from 3 to 4.

**Key Questions For Authors:**

Please, see weaknesses above.

**Limitations:**

Yes.

**Strengths And Weaknesses:**

### Strengths

- The covering number argument is simple. Although noise is treated as input to the hypothesis class, this allows comparing with naive noise injection. By defining a semi-metric that collapses noise configurations related by the transformation group T, the authors give a clean and rigorous explanation for why equivariant noise generalizes better. Propositions 4.2 and 4.3 make this concrete: larger symmetry group means fewer covering points, and permutation invariance alone gives a C! reduction factor.

- The experiments cover four task types (node, link, subgraph, and graph prediction), which is broader than most papers in this area. The scalability results in Table 7 are also convincing: ENGNN uses roughly 10% of the resources required by subgraph GNNs.

- Code is available for review.

---

### Weaknesses

**Theory.** My main concerns are related to the expressivity results for noise-augmented graphs.

In my understanding, a key aspect of using noise to boost expressivity is that permutation invariance is then achieved only in distribution, as discussed in (Abboud, 2019). The paper does not acknowledge this point. As a result, several notions and results appear somewhat imprecise or informal.

In particular, the paper builds upon on the connection between universal approximation and isomorphism discrimination studies by Chen et al. (2019). The paper restate their definition of Graph-Isomorphism-discriminating as “... for any two non-isomorphic graphs $G_1, G_2$, there exists a parameterization $\theta$ such that the GNN outputs are different: $f_\theta(G1) \neq f_\theta(G2)$.” However, Chen et al. (2019) restrict their results to permutation-invariant functions. Thus, for a model to be G-Iso-discriminating (and universal approximator), it must also satisfy that $f_\theta(G_1) = f_\theta(G_2)$ whenever $G_1 \cong G_2$ (they are isomorphic). Noise-augmented graphs should therefore be treated as random variables, and a clear notion of universal approximation should be stated under this stochastic formulation.

Instead, the paper treats noise as deterministic input. This makes some results rather trivial, such as the invariance property stated in Theorem 4.4.

For the universal expressivity result (Theorem 4.5), the statement requires node-level noise to be distinct, i.e., $z_i \neq z_j$ for all distinct nodes $i \neq j$, but then says:
$$
ENGNN(G, Z_1) \neq ENGNN(H, Z_2),  \forall Z_1, Z_2 \in \mathcal{Z}
$$
This is clearly contradictory, as the result is stated to hold for all noise matrices, including configurations that violate the distinctness assumption.

I also have the following question:

- Could the authors elaborate on why requiring within-graph node-level random features to be distinct suffices to obtain universal approximation with the proposed architecture? For instance, consider
  - G = two disjoint 3-cycles
  - H = a single 6-cycle

Assume that all nodes in both graphs have identical input features $X$. Now assume that you have $Z_G \in \mathbb{R}^{6 \times 1}$ such that each row is different (${Z_G}_i \neq {Z_G}_j$ for all $i \neq j$), and $Z_H=Z_G$. How does your method (aggregation schemes) ensure that $ENGNN(G, Z_G) \neq ENGNN(H, Z_H)$?

Finally, I believe you should also make it explicit that universality holds for graph spaces of a fixed maximum size $n$, no? I am saying it because universal approximation usually requires the approximation to hold simultaneously over the entire domain.

**Presentation.** Overall, I found the paper relatively difficult to follow. One issue is that the architecture is described very generically in the main text, with key details deferred to the appendix. As a result, Section 4.2, which is supposed to introduce the proposed method, remains very non-informative.

In addition, the imprecise nature of some notions and results (as discussed above) also contributes to hinder readability, including the proofs.

Also, I think some choices are not clearly motivated. For instance, it was unclear to my why the paper assumes $Z_i^{(k)} \in L \times C $ for all $k>1$.

**Experiments.**

- A notable omission is the expressivity evaluation on BREC benchmarks (Wang & Zhang, 2024). How does the proposed method perform on BREC datasets?

- Also, the evaluation omits Graph Transformers (GPS, Graphormer), which are competitive on ZINC and MOLHIV and use positional encodings, which is conceptually close to ENGNN --- in the sense it combines node features with additional information. Without this comparison it's hard to tell if ENGNN is actually advancing beyond what practitioners already use.

- In Table 10, why are the results for GPSE on MUTAG and NCI1 missing?

- Finally, could the authors comment on the tightness and practical relevance of the derived generalization bounds? To what extent do these bounds meaningfully explain the empirical results reported in the tables?

---

> ### Author Rebuttal · Authors · 2026-03-31
>
> We thank the reviewer for the thoughtful and detailed feedback. We address some of your concerns and questions below due to length limit:
> * W1. The paper does not acknowledge that when using noise to boost expressivity, permutation invariance is only achieved in distribution.
>
> Our method involves two distinct types of invariance:
> 1. Channel Invariance: For a fixed graph, our model's output is invariant/equivariant to transformations of the noise channels (e.g., permutations or rotations of the noise features). This is our primary theoretical focus.
> 2. Node Permutation Invariance: For a noise distribution that is invariant to node reordering, the expectation of our model's output is invariant to node permutations.
>
> As the second type is a well-established property in existing noise GNN literature, we focused our discussion on the former. We will explicitly clarify these two layers of invariance in the revision.
> * W1.1 The paper relies on Chen et al. (2019) for the connection between universality and isomorphism, but their results are restricted to permutation-invariant functions.
>
> We acknowledge the gap between our initial theoretical conclusion and the strict requirement for permutation-invariant functions in the framework of Chen et al. (2019). This can be resolved by considering the expectation of the ENGNN output over the noise distribution. Specifically, let $f_{\\theta}(G) = E_{Z \\sim \\mathcal{Z}} [ENGNN_{\\theta}(G, Z)]$. We demonstrate that $f_{\\theta}$ satisfies the two conditions required by Chen et al. (2019):
>
> 1. $f_{\\theta}(G_1) = f_{\\theta}(G_2)$ whenever $G_1 \\cong G_2$.
>
> 2. For all $G_1 \ncong G_2$, there exists a parameter $\theta$ such that $f_{\theta}(G_1) \neq f_{\theta}(G_2)$.
>
> Proof: Let $S_1 = \\{ENGNN_{\\theta}(G_1, Z) | Z \\in \\mathcal{Z}\\}$ and $S_2 = \\{ENGNN_{\\theta}(G_2, Z) |Z \\in \\mathcal{Z}\\}$ be the sets of potential model outputs for each graph across all possible noise realizations.As shown in Equation 8 of our Theorem 4.5, there exists a configuration $\theta'$ such that $S_1 \cap S_2 = \emptyset$. We can therefore define a function $g$ that maps these disjoint sets to distinct real numbers: g(x) = 1 if x in S1 and  g(x)=2 if x in S2. By incorporating g into the output layer of the ENGNN, we obtain a parameter $\theta$ such that $f_{\theta}(G_1) = 1 \neq 2 = f_{\theta}(G_2)$, satisfying the discrimination requirement.
>
> * W1.3 Theorem 4.5 requires node-level noise to be distinct, yet claims $ENGNN(G_1, Z_1) \neq ENGNN(G_2, Z_2)$ for all $Z_1, Z_2 \in \mathcal{Z}$, which seems contradictory.
>
> The contradiction is resolved by the definition of the noise space $\mathcal{Z}$. As in the text of Theorem 4.5, we assume noise configurations satisfy the "distinctness" and the requirements of the equivariant aggregator. These constraints only exclude a low-dimensional manifold of measure zero from the total space. In the revision, we will explicitly define $\mathcal{Z}$ as the set of all noise configurations that satisfy constraints.
>
> * W2.1 How does your method ensure $ENGNN(G, Z_G) \neq ENGNN(H, Z_H)$ for two disjoint 3-cycles vs. a single 6-cycle when node features are identical and noise $Z$ is distinct for all nodes?
>
> For this case, our ENGNN can first mapping each difference noise number to 6-d one hot vector indicating node identity, then sum neighbor's one hot vector in each layer to update feature. After 3 layers, for H,  all elements in feature vector are non-zeros, but for G each node's feature vector has at most 3 non zero elements.
>
> For general case, noise assigns node order, than GNN can simulate ordinary NN with adjacency matrix and featureu matrix as input.
>
> * W4.1 Add expressivity evaluation on BREC benchmarks (Wang & Zhang, 2024).
>
> Our ENGNN-O solves much more cases than Graphormer:
> |Model| Basic (60)|Regular (140)|Extension (100)|CFI (100)|
> |-|-|-|-|-|
> |ENGNN-O|60|70|98|14|
> |Graphormer|16|12|41| 10|
>
> * W4.2 Evaluation omits Graph Transformers... which are conceptually close to ENGNN in their use of positional encodings.
>
> ENGNN achieves performance competitive to these models:
> |Model|ZINC (MAE ↓)|MOLHIV (ROC-AUC ↑)|
> |-|-|-|
> |Graphormer|0.122±0.006 |80.51±0.53
> |GPS|0.070±0.004|78.80±1.01|
> |ENGNN-P|0.091±0.005|78.51±0.86|
> |ENGNN-O|0.070±0.006|78.63±0.93|
>
> * W4.4 The practical relevance of the derived generalization bounds.
>
> The practical usage is: the more equivariance we use in ENGNN, the smaller generation error bound we can get. For noise MPNN, it is invariant to group with only identity mapping. For ENGNN-P, it is invariant to a larger permutation group. ENGNN-O is invariant to an even larger orthogonal group. It is consistent with our experiment results: in general ENGNN-O performs better than ENGNN-P, and vanilla MPNN with noise performs worst in general.
>
> [1] On learning sets of symmetric elements. ICML 2020.
> [2] Scalars are universal: Equivariant machine learning, structured like classical physics. NeurIPS 2021.

---

> > ### Author Rebuttal · Reviewer_L1SF · 2026-04-03
> >
> > I would like to thank the authors for their responses and for running the additional experiments I suggested.
> >
> > I have a few remaining questions/comments:
> >
> > 1. It is still unclear to me how the proposed model distinguishes between two disjoint 3-cycles and a single 6-cycle when their noise matrices are identical. I would appreciate a more detailed explanation of the underlying procedure. I didn't get what the authors mean by “difference noise number” and that “noise assigns node order”. If the noise matrices are the same for both graphs, wouldn’t the induced node identities also coincide?
> >
> > 2. Could the authors clarify how the model was evaluated on the BREC dataset? The original benchmark involves node permutations, and it is not clear how this was handled in your setup. Did you consider the expected version of ENGNN-O via Monte Carlo approximation? If so, how many samples were used? Were predictions thresholded to ensure (node) permutation invariance? Finally, could you specify whether ENGNN was trained or evaluated with random parameters? Given the claimed universality of ENGNN, should the model not, in principle, distinguish all graph pairs in the benchmark?
> >
> > I would also appreciate the authors’ clarification on a few points unaddressed from my initial review, namely:
> >
> > - In Table 10, why are the results for GPSE on MUTAG and NCI1 missing?
> > - Finally, I believe you should also make it explicit that universality holds for graph spaces of a fixed maximum size $n$, no? I am saying it because universal approximation usually requires the approximation to hold simultaneously over the entire domain.
> > - Also, I think some choices are not clearly motivated. For instance, it was unclear to my why the paper assumes [...]

---

> > > ### Author Response · Authors · 2026-04-07
> > >
> > > We are glad that our previous response addressed some of your concerns. Due to the reply length limit, we were unable to address all points initially; we provide those remaining explanations below.
> > >
> > > AQ1. It is still unclear to me how the proposed model distinguishes between two disjoint 3-cycles and a single 6-cycle when their noise matrices are identical.
> > >
> > > The noise does not need to be unique across different graphs; it only needs to distinguish nodes within a single graph instance. Even if the set of noise values is identical for two different graphs, the structural connectivity will propagate that noise differently during message passing.
> > >
> > > Assume two 3-cycles has 6 nodes v1,v2,v3,v4,v5,v6, 6-cycle u1,u2,u3,u4,u5,u6, with bidirectional edge v1-v2-v3-v1, v4-v5-v6-v4, u1-u2-u3-u4-u5-u6-u1.
> > >
> > > Initially, assume v1 and u1 have the same noise 0.1, v2/u2 -0.2, v3/u3 0.3, v4/u4 -0.1, v5/u5 1.0, v6/u6 -0.5.
> > >
> > > As nodes noise are different, there exists a mlp to map noise value to a one-hot vector. Then the node representation is
> > >
> > > v1/u1 [1,0,0,0,0,0], v2/u2 [0,1,0,0,0,0], v3/u3 [0,0,1,0,0,0], v4/u4 [0,0,0,1,0,0], v5/u5 [0,0,0,0,1,0], v6/u6 [0,0,0,0,0,1].
> > >
> > > In each message passing layer, we let each node to sum neighbors' feature vector. After first layer.
> > >
> > > v1 [1,1,1,0,0,0], v2 [1,1,1,0,0,0], v3 [1,1,1,0,0,0], v4 [0,0,0,1,1,1], v5 [0,0,0,1,1,1], v6 [0,0,0,1,1,1].
> > >
> > > u1 [1,1,0,0,0,1], u2 [1,1,1,0,0,0], u3 [0,1,1,1,0,0], u4 [0,0,1,1,1,0], u5 [0,0,0,1,1,1], u6 [1,0,0,0,1,1].
> > >
> > > We let the pooling layer count the number of nodes with feature [0,1,1,1,0,0], then two cycle produces 0, 6-cycle produces 1. Then two graph are distinguished.
> > >
> > > The "noise assigns node order" simply means that the unique noise values act as temporary node identities. With these identities, a single layer of message passing effectively encodes the edge list into the node features.
> > >
> > > AQ2. Could the authors clarify how the model was evaluated on the BREC dataset?
> > >
> > > We followed the standard routine for random GNN provided in the BREC benchmark (as detailed in the end of Section 5 of the BREC paper[1]):
> > >
> > > Permutation Invariance: We do not use an expected version via Monte Carlo approximation. Instead, we use the prediction produced by ENGNN-O for each evaluation instance. BREC performs 100 evaluations with different graph permutations; in each forward pass, our ENGNN-O uses resampled noise.
> > >
> > > Output: We pass the raw prediction from the linear head directly to the BREC evaluation routine.
> > >
> > > Training: ENGNN-O is trained using the loss function defined in Equation 1 of the BREC paper[1].
> > >
> > > AQ2-2. Given the claimed universality of ENGNN, should the model not, in principle, distinguish all graph pairs in the benchmark?
> > >
> > > The gap between theoretical universality and empirical performance is a known challenge in expressive GNNs. In the BREC benchmark, even the most expressive models typically solve ~70% of cases; ENGNN-O solves ~60%, while a vanilla MPNN solves 0%.
> > >
> > > The failures often stem from numerical stability and prediction variance. Since BREC measures the difference between graph predictions relative to the variance of the same graph, if the randomness-induced variance is too high, the test considers the case "failed" even if the model produces different prediction for two graphs.
> > >
> > > AQ3. In Table 10, why are the results for GPSE on MUTAG and NCI1 missing?
> > >
> > > The omission of GPSE results for NCI1 and MUTAG was a typo. Using the default setting of GPSE (pretrained on molpcba), the auroc results are: NCI1: GPSE: 0.826±0.010 vs. ENGNN-O: 0.902±0.022; MUTAG: GPSE: 0.960±0.019 vs. ENGNN-O: 0.990±0.014. ENGNN-O consistently outperforms GPSE on both datasets.
> > >
> > > AQ4. Finally, I believe you should also make it explicit that universality holds for graph spaces of a fixed maximum size, no? I am saying it because universal approximation usually requires the approximation to hold simultaneously over the entire domain.
> > >
> > > We agree. We will explicitly state in the revision that for any fixed maximum number of nodes n > 1, there exists an ENGNN architecture that achieves universal expressivity over that graph space. As n increases, the required model capacity may also increase.
> > >
> > > AQ5. Also, I think some choices are not clearly motivated. For instance, it was unclear to my why the paper assumes $Z_{i}^{(k)} \in \mathbb{R}^{L \times C}$ for $k > 1$.
> > >
> > > This choice is necessitated by the requirements of maintaining equivariance across layers. Since the input noise Z consists of C channels, we require the internal equivariant representations to transform (e.g., permute) in a manner consistent with those C channels, so C dimension is needed. The additional dimension $L$ acts as a hidden feature dimension, allowing us to apply learnable linear transformations and non-linearities within the equivariant stream. This design is a common practice in geometric deep learning[2].
> > >
> > > [1] An Empirical Study of Realized GNN Expressiveness. ICML 2024.
> > > [2] E(n) Equivariant Graph Neural Networks. ICML 2021.

---

### Decision · Program_Chairs · 2026-04-30

**Decision:**

Accept (spotlight)

**Comment:**

In this paper, the authors introduce expressive variants of GNNs based on random features, used in an equivariant manner. The reviewers praised the idea, adressing a long-standing problem in GNN theory and application (that noise help generalization but breaks equivariance). They asked some clarification about the theory and empirical experiments, to which the authors answered in detail. The overall assessment is positive.